# Acquired IFNγ resistance impairs anti-tumor immunity and gives rise to T-cell-resistant melanoma lesions

Antje Sucker[1,2], Fang Zhao[1,2], Natalia Pieper[1,2], Christina Heeke[1,2], Raffaela Maltaner[1,2], Nadine Stadtler[1,2], Birgit Real[1,2], Nicola Bielefeld[1,2], Sebastian Howe[3], Benjamin Weide[4], Ralf Gutzmer[5], Jochen Utikal[6], Carmen Loquai[7], Helen Gogas[8], Ludger Klein-Hitpass[9], Michael Zeschnigk[10], Astrid M. Westendorf[11], Mirko Trilling[3], Susanne Horn[1,2], Bastian Schilling[1,2,12], Dirk Schadendorf[1,2], Klaus G. Griewank[1,2] & Annette Paschen[1,2]

Melanoma treatment has been revolutionized by antibody-based immunotherapies. IFNγ secretion by CD8+ T cells is critical for therapy efficacy having anti-proliferative and pro-apoptotic effects on tumour cells. Our study demonstrates a genetic evolution of IFNγ resistance in different melanoma patient models. Chromosomal alterations and subsequent inactivating mutations in genes of the IFNγ signalling cascade, most often *JAK1* or *JAK2*, protect melanoma cells from anti-tumour IFNγ activity. JAK1/2 mutants further evolve into T-cell-resistant HLA class I-negative lesions with genes involved in antigen presentation silenced and no longer inducible by IFNγ. Allelic *JAK1/2* losses predisposing to IFNγ resistance development are frequent in melanoma. Subclones harbouring inactivating mutations emerge under various immunotherapies but are also detectable in pre-treatment biopsies. Our data demonstrate that JAK1/2 deficiency protects melanoma from anti-tumour IFNγ activity and results in T-cell-resistant HLA class I-negative lesions. Screening for mechanisms of IFNγ resistance should be considered in therapeutic decision-making.

[1] Department of Dermatology, University Hospital Essen, University Duisburg-Essen, 45122 Essen, Germany. [2] German Cancer Consortium (DKTK), partner site Essen/Düsseldorf, 45122 Essen, Germany. [3] Institute of Virology, University Hospital Essen, University Duisburg-Essen, 45122 Essen, Germany. [4] Division of Dermatooncology, Department of Dermatology, University Medical Center Tübingen, 72076 Tübingen, Germany. [5] Department of Dermatology and Allergy, Skin Cancer Center Hannover, Hannover Medical School, 30625 Hannover, Germany. [6] German Cancer Research Center (DKFZ), Skin Cancer Unit, Heidelberg and University Medical Center Mannheim, Department of Dermatology, Venereology and Allergology, Ruprecht-Karl University of Heidelberg, 68167 Mannheim, Germany. [7] Skin Cancer Center, Department of Dermatology, University of Mainz Medical Center, 55131 Mainz, Germany. [8] First Department of Medicine,National and Kapodistrian University of Athens, 11527 Athens, Greece. [9] Institute of Cell Biology, University Hospital Essen, University of Duisburg-Essen, 45122 Essen, Germany. [10] Institute of Human Genetics, University Hospital Essen, University Duisburg-Essen, West German Cancer Center and the German Cancer Consortium (DKTK), 45122 Essen, Germany. [11] Institute of Medical Microbiology, University Hospital Essen, University of Duisburg-Essen, 45122 Essen, Germany. [12] Department of Dermatology, Venereology and Allergology, University Hospital Würzburg, 97080 Würzburg, Germany. Correspondence and requests for materials should be addressed to A.P. (email: annette.paschen@uk-essen.de).

Understanding the mechanisms of T-cell inhibition by melanoma cells allowed for the development of new agents with considerable activity against metastatic disease including antibodies targeting the PD-L1/PD1 axis. PD-L1 expressed on melanoma cells binds its inhibitory PD1 receptor on cytotoxic CD8[+] T lymphocytes generating a checkpoint signal dampening the T cell's effector function[1]. Release from checkpoint blockade by treatment with anti-PD1 antibodies yields clinical benefit in a substantial proportion of melanoma patients, experiencing durable disease stabilization, tumour regression as well as complete remission[2–4]. Response to anti-PD1 therapy is strongly associated with the expression of its ligand on melanoma cells and the presence of CD8[+] T cells in the margin or center of metastatic lesions[5]. How T cells mediate disease stabilization or regression of bulky tumour masses remained unclear so far.

Upon activation by cognate HLA class I antigen complexes, T cells release cytolytic granules, containing perforins and granzymes, onto their target cells and secrete interferon (IFN)γ acting on cells in the microenvironment[6]. Perforin/granzyme-mediated killing and induction of apoptosis by death receptor engagement have long been considered the major anti-tumour effector mechanisms of CD8[+] T cells. Accordingly, expression of cytolytic markers in pretreatment melanoma biopsies was found to be significantly associated with clinical benefit to antibodies targeting the T-cell checkpoint CTLA-4 (ref. 7). But evidence from different in vivo studies suggests that the anti-proliferative and pro-apoptotic activity of IFNγ on melanoma cells contributes essentially to the efficacy of T-cell-mediated anti-tumour immunity.

IFNγ binds to the heterodimeric IFNGR1/IFNGR2 receptor complex, leading to the activation of the receptor-associated kinases JAK1 and JAK2 that in turn phosphorylate STAT1. Phosphorylated STAT1 homodimers activate transcription of primary response genes including the transcriptional activator IRF1 that in turn coordinates the expression of secondary response genes[8]. Activation of the JAK1/2-STAT1-IRF1 signalling cascade in melanoma cells as well as other tumour cells can induce growth arrest and death via different pathways[9–12]. Recently, it was demonstrated that adoptively transferred tumour antigen-specific CD8[+] T cells infiltrating B16 melanoma lesions at low numbers arrested the growth of several times higher numbers of tumour cells in an IFNγ-dependent manner[11]. Furthermore, T-cell-derived IFNγ in combination with tumour-necrosis factor (TNF)α was found to be essential also for in vivo induction of tumour-cell senescence abrogating disease progression in a pancreatic tumour model[13,14].

Based on this knowledge, we postulated that melanoma cells from patients responding to immunotherapy should be sensitive to the anti-proliferative and pro-apoptotic effects of IFNγ and that continuous cytokine exposure should select for the outgrowth of IFNγ-resistant tumour subclones. Here we demonstrate that IFNγ-resistant melanoma clones with inactivating JAK1/JAK2 mutations frequently evolve in patients receiving different types of immunotherapy. IFNγ-resistant tumour cells are protected from cytokine-induced growth inhibition and apoptosis. Additionally, JAK1/JAK2-deficient lesions become T-cell-resistant by silencing HLA class I antigen presentation, which can no longer be restored by IFNγ signalling. Our findings suggest sequential screening of tumour biopsies for genetic defects in the IFNγ signalling cascade will aid therapeutic decision-making in patients with advanced melanoma.

## Results

**Acquired mutations in genes of the IFNγ signalling pathway.** Assuming that the growth-inhibitory and pro-apoptotic activity of T-cell-derived IFNγ acts selectively on tumour cells, the evolution of genetic variants in melanoma with impaired cytokine signalling was explored. In a first step, we evaluated available exome data of 46 melanoma cell lines[15], established from metastases of different patients in our laboratory, for aberrations in IFNGR1, IFNGR2, JAK1, JAK2, STAT1 and IRF1. Mutations in JAK1 ($n=3$), JAK2 ($n=1$) and STAT1 ($n=1$) were detected in 5 out of the 46 cell lines (Table 1). By Sanger sequencing we confirmed the mutations on freshly isolated DNA from the respective cell lines Ma-Mel-36, Ma-Mel-53, Ma-Mel-54a, Ma-Mel-85 and Ma-Mel-102. Independent of existing exome data, Sanger sequencing revealed a JAK1 mutation in a cell line from melanoma patient Ma-Mel-61 (Table 1). The specific mutations present in the cell lines were also detected in situ in corresponding tumour tissue, with the exception of metastasis Ma-Mel-54a. As shown in Table 1, targeted sequencing revealed a homozygous status for the mutant allele in three of the six cell lines (Ma-Mel-54a, Ma-Mel-61g, Ma-Mel-102). To determine whether these mutations functionally impaired IFNγ signalling, the cell lines were treated with recombinant IFNγ for 48 h followed by protein expression analyses of pathway components and downstream targets.

Despite a STAT1 (c.947C > T) mutation frequency of approximately ~100%, Ma-Mel-102 cells still showed a slight induction of pSTAT1 and IRF1 in the presence of IFNγ (Supplementary Fig. 1a–c). However, the signals were weak and detectable only after long-term exposure of X-ray films, suggesting that the S316L exchange, located between the coiled-coil and DNA binding domains, strongly decreased STAT1 protein stability without necessarily leading to its complete inactivation. Parallel analyses on Ma-Mel-85 cells revealed a strong IFNγ pathway activation compared with Ma-Mel-102 cells, accompanied with an elevated surface expression of CD54, HLA class I and PD-L1 (Supplementary Fig. 1b–d). This is in line with a JAK1 (c.1548C > A) mutation frequency of only 50% (Supplementary Fig. 1a), suggesting that wild-type JAK1 was still active in Ma-Mel-85 cells. As expected, IFNγ signalling was detected also in Ma-Mel-53 cells showing a JAK1 (c.2338G > A) mutation frequency of only 24% (Supplementary Fig. 1a–c).

**Table 1 | Acquired mutations in genes of the IFNγ signalling pathway.**

| Patient | Metastasis excision | Disease stage | Cell line | Mutated gene | Allelic status | DNA | Protein | Mutation in tissue |
|---|---|---|---|---|---|---|---|---|
| Ma-Mel-36 | 05/2000 | IV | Ma-Mel-36 | JAK1 | het* | c.843C > A | p.Y281* | Yes |
| Ma-Mel-53 | 03/2002 | IV | Ma-Mel-53 | JAK1 | het | c.2338G > A | p.G780R | Yes |
| Ma-Mel-85 | 03/2004 | IV | Ma-Mel-85 | JAK1 | het | c.1548C > A | p.F516L | Yes |
| Ma-Mel-54 | 09/2002 | IV | Ma-Mel-54a | JAK2 | hom | c.2876A > C | p.Q959P | n.d. |
| Ma-Mel-102 | 11/2004 | III | Ma-Mel-102 | STAT1 | hom | c.947C > T | p.S316L | Yes |
| Ma-Mel-61 | 05/2005 | IV | Ma-Mel-61g | JAK1 | hom | c.1798G > T | p.G600W | Yes |

n.d., not detectable.
*Consists in equal parts of homozygous wild-type and homozygous mutant subpopulations.

**IFNγ resistance protects from cytokine-induced cell death.** Targeted sequencing pointed to a heterozygous *JAK1* mutation in cell line Ma-Mel (Table 1) that was established from a cutaneous patient metastasis (Fig. 1a). However, when treated with IFNγ, analyses of HLA-DR, HLA class I and PD-L1 surface expression on Ma-Mel-36 cells (hereafter referred to as Ma-Mel-36_bulk cells) demonstrated that the cell line consisted in equal parts of IFNγ-sensitive and IFNγ-resistant subpopulations (Fig. 1b). Both subpopulations were sorted from IFNγ-treated Ma-Mel-36_bulk cells based on their different

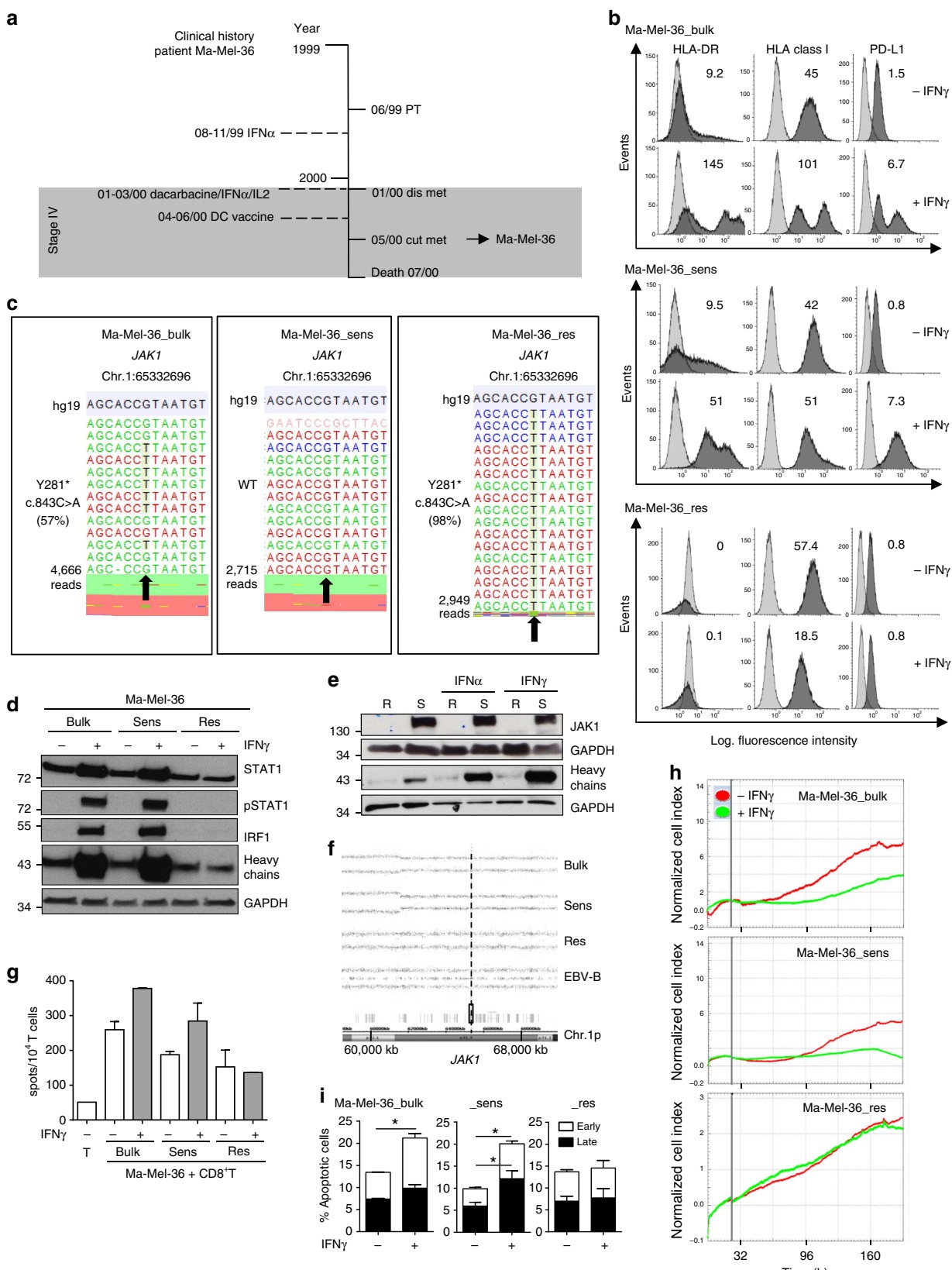

HLA-DR expression profiles. The IFNγ-sensitive subpopulation Ma-Mel-36_sens strongly upregulated HLA-DR and PD-L1 surface expression in response to cytokine treatment, whereas the IFNγ-resistant subpopulation Ma-Mel-36_res remained HLA-DR-negative and PD-L1-low under these conditions (Fig. 1b). By targeted sequencing we detected the JAK1 mutation (c.843C>A) encoding a truncated non-functional JAK1-Y281* variant in ∼60 and 100% of Ma-Mel-36_bulk and Ma-Mel-36_res cells, respectively, but not in Ma-Mel-36_sens cells (Fig. 1c). Accordingly, IFNγ treatment resulted in pSTAT1 and IRF1 detection in lysates from Ma-Mel-36_bulk and Ma-Mel-36_sens cells but not from Ma-Mel-36_res cells (Fig. 1d). In line with the sequencing results, JAK1 protein expression was detected only in Ma-Mel-36_sens and not in Ma-Mel-36_res cells (Fig. 1e).

Assuming that JAK1 deficiency in these cells was due to a gene mutation and concurrent allelic loss, we performed single-nucleotide polymorphism (SNP) array analyses on DNA obtained from the three tumour cell populations and autologous Epstein-Barr virus (EBV)-transformed B cells as a control. The same partial deletion on chromosome 1p, encompassing the region 1p36.3-1p13.1 (Chr.1:854,277-116,804,754) including the JAK1 gene mapping at Chr.1p31.3 was detected in Ma-Mel-36_bulk, Ma-Mel-36_sens and Ma-Mel-36_res cells (Fig. 1f). This result demonstrated that the allelic JAK1 loss occurred early in the course of disease and that a subsequent c.843C>A mutation in the remaining JAK1 allele generated the JAK1-deficient Ma-Mel-36_res subpopulation. Consistently, JAK1 reconstitution by transient transfection of Ma-Mel-36_res cells with a JAK1 expression plasmid restored IFNγ signalling (Supplementary Fig. 2a–c).

As shown in Fig. 1a, metastasis Ma-Mel-36 developed after the patient had been treated with recombinant IFNα and a combination of dacarbacine/IFNα/interleukin (IL) 2, suggesting activated tumour-reactive T cells selectively enriched the IFNγ-resistant cell subpopulation. Indeed, peripheral blood CD8+ T cells from patient Ma-Mel-36 secreted IFNγ in the presence of autologous melanoma cells, as determined by ELISpot assay (Fig. 1g). Pretreatment of tumour cells with IFNγ slightly enhanced the activation of CD8+ T cells by Ma-Mel-36_bulk and Ma-Mel-36_sens cells, whereas the T-cell-stimulatory capacity of Ma-Mel-36_res cells was not affected. Furthermore, impedance-based measurement of real-time proliferation in the xCELLigence system revealed a negative impact of IFNγ on the expansion of Ma-Mel-36_bulk and Ma-Mel-36_sens cells, while Ma-Mel-36_res cells efficiently proliferated (Fig. 1h). This was measurable also in terms of cell numbers: a considerable reduction in Ma-Mel-36_bulk and in particular Ma-Mel-36_sens cells was noted in the presence of IFNγ, due to an increase in apoptosis (Fig. 1i and Supplementary Fig. 2d,e). In contrast, cell numbers and spontaneous apoptosis of IFNγ-resistant Ma-Mel-36_res cells remained unaffected under these conditions (Fig. 1i and Supplementary Fig. 2d,e).

As shown in Fig. 1e, Ma-Mel-36_sens cells responded to IFNα treatment whereas JAK1-deficient Ma-Mel-36_res cells were also resistant to type I IFN. Considering patient Ma-Mel-36 received IFNα-based therapies before the development of resistant lesions, we hypothesized that type I IFN signalling by affecting cell survival might have contributed to the enrichment of JAK1-deficient cells. However, in contrast to IFNγ, IFNα treatment did not affect the survival of Ma-Mel-36_sens cells (Supplementary Fig. 2f).

**JAK2 deficiency blocks HLA class I upregulation by IFNγ.** By Sanger sequencing we found a JAK2 c.2876A>C exchange to be present in Ma-Mel-54a cells that, however, could not be detected in the corresponding tumour tissue (Table 1). To demonstrate that, in fact, the specific genetic alteration was acquired in the course of disease we sequenced DNA from a second cutaneous lesion (Ma-Mel-54b) of the patient, obtained one month after excision of metastasis Ma-Mel-54a (Fig. 2a). Indeed, tumour tissue Ma-Mel-54b and the corresponding cell line harboured the JAK2 mutation already present in Ma-Mel-54a cells (Fig. 2b). Both cell lines showed a JAK2 (c.2876A>C) mutation frequency of 100%, resulting in a Q959P exchange in the functionally important JAK2 JH1 kinase domain[16]. As shown in Fig. 2c, mutant JAK2-Q959P was no longer detectable by western blot. Accordingly, IFNγ signalling was completely abrogated in both cell lines, which no longer showed CD54, HLA class I and PD-L1 upregulation in response to cytokine treatment (Fig. 2d,e).

Assuming that JAK2 deficiency of Ma-Mel-54 cells was caused by the co-occurrence of a JAK2 gene mutation and allelic JAK2 loss, we performed SNP array analyses on DNA obtained from the two cell lines and autologous peripheral blood cells as a constitutive, normal control to detect aberrations of chromosome 9p to which the JAK2 gene maps at Chr.9p24.1. The same deletion on chromosome 9p, encompassing the region 9p24.3–p13.2 (Chr.9:203,861-37,578,327) was detected in Ma-Mel-54a and Ma-Mel-54b cells (Fig. 2f), demonstrating the common origin of JAK2 deficiency in both metastases. As shown in Fig. 2g, IFNγ sensitivity of Ma-Mel-54a cells was restored upon transient JAK2 re-expression as indicated by the induction of signalling pathway components. Furthermore, Ma-Mel-54a-JAK2

**Figure 1 | Protection from cytokine-induced cell death by acquired IFNγ resistance.** (**a**) Clinical history of patient Ma-Mel-36. Vertical line, time axis; left, therapeutic regimens; right, primary tumour (PT)/metastases development; arrow indicates cell line established from metastasis Ma-Mel-36; grey box, stage IV disease. (**b**) IFNγ-sensitive Ma-Mel-36_sens and IFNγ-resistant Ma-Mel-36_res cells sorted from IFNγ-treated (48 h) Ma-Mel-36_bulk cells based on their HLA-DR expression profile. Surface expression of indicated proteins measured by flow cytometry. Representative data from n=3 independent experiments. (**c**) JAK1 mutation defined by targeted sequencing on DNA from Ma-Mel-36_bulk and Ma-Mel-36_res cells. Plots of aligned sequencing reads in the location where the JAK1 c.843C>A, p.Y281* mutation was identified, arrows highlight mutation or corresponding wild-type (WT) site. Number of sequencing reads notated on the left; %, frequency of mutation in reads. (**d**) Melanoma cells analysed by western blot for expression of STAT1, pSTAT1, IRF1 and HLA class I heavy chains after IFNγ treatment (48 h); GAPDH, loading control. Representative data from n=2 independent experiments. (**e**) Ma-Mel-36_sens (S) and Ma-Mel-36_res (R) cells analysed for protein expression after IFNα and IFNγ treatment (48 h). Representative data from n=2 independent experiments. (**f**) SNP results given as allelic distribution of chromosome 1p shown for DNA obtained from Ma-Mel-36_bulk, Ma-Mel-36_sens, Ma-Mel-36_res and autologous Epstein-Barr virus-transformed B cells as a control. Loss of one chromosomal allele in the region 1p36.3-1p13.1 (Chr.1:854,277-116,804,754; hg19) in all Ma-Mel-36 cell populations. Dashed line indicates JAK1 location at Chr.1p31.3. (**g**) IFNγ release by autologous CD8+ T cells in the presence of IFNγ-treated (24 h) Ma-Mel-36 cell populations measured by ELISpot assay. Mean values (+s.e.m.) from n=2 measurements. (**h**) Real-time proliferation of Ma-Mel-36 cell populations in the presence/absence of IFNγ. Bold grey vertical lines indicate addition of IFNγ. Representative data from n=3 independent experiments. (**i**) IFNγ-induced (7 days) apoptosis in Ma-Mel-36 cell populations determined by AnnexinV/PI staining. Percentage of early (AnnexinV+/PI−) and late apoptotic (AnnexinV+/PI+) cells depicted. Mean values (+s.e.m.) from n=3 independent experiments. Only statistical significant differences defined by paired Student's t-test are indicated, *P<0.05.

transfectants proved to be sensitive towards the anti-proliferative activity of IFNγ in contrast to non-transfected Ma-Mel-54a cells (Fig. 2h).

Interestingly, western blot analyses suggested a lack of HLA class I heavy chain expression in Ma-Mel-54a compared with Ma-Mel-54b cells (Fig. 2d). Indeed, Ma-Mel-54a cells only weakly expressed HLA class I surface molecules (Fig. 2e). By quantitative reverse transcription–PCR we demonstrated low level expression of specific messenger RNAs (mRNAs) involved in antigen presentation in Ma-Mel-54a cells (Fig. 2i). Expression of

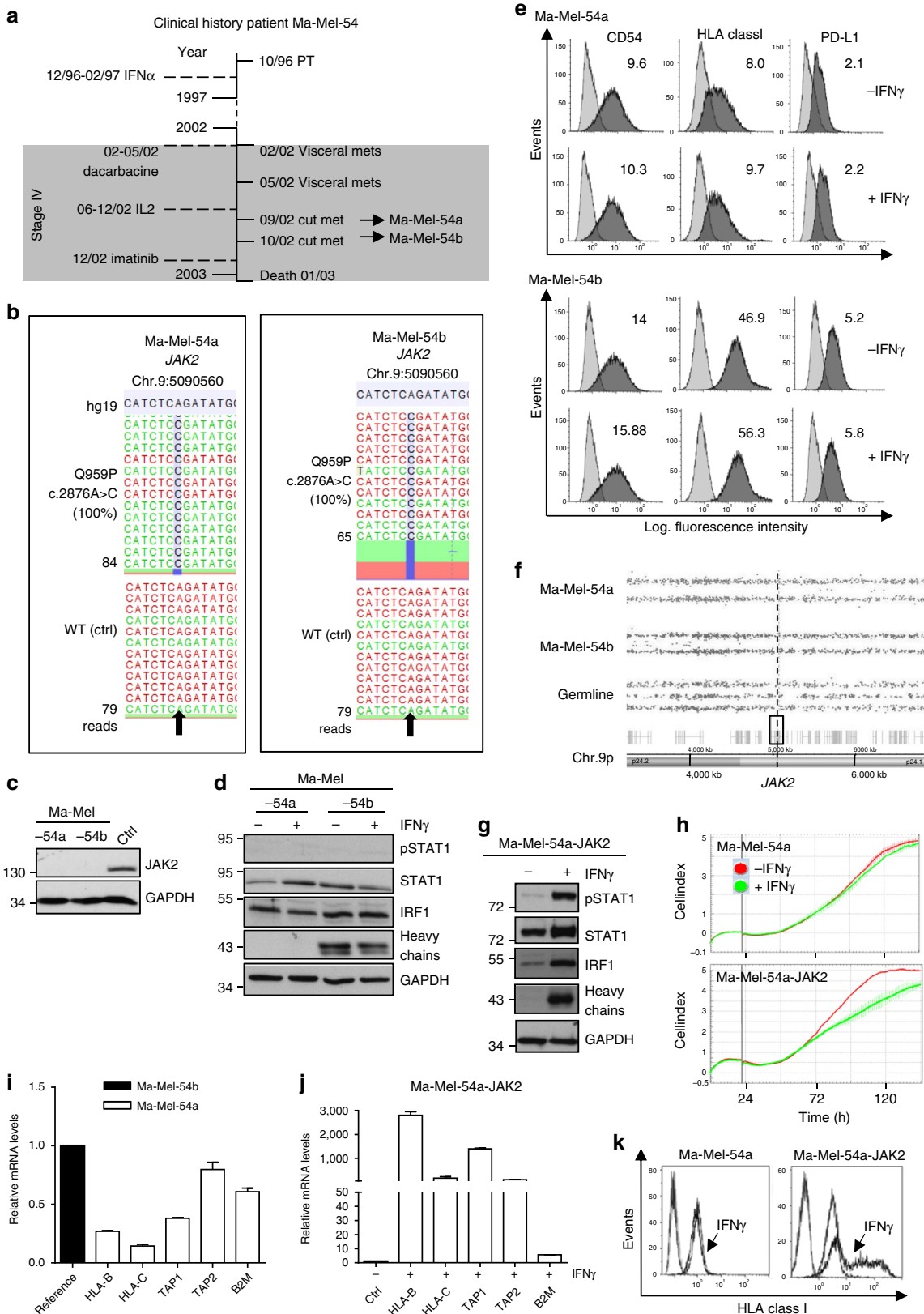

*HLA-B, HLA-C, TAP1, TAP2* and *B2M* genes as well as HLA class I surface molecules were strongly upregulated solely by IFNγ-treated Ma-Mel-54a-JAK2 transfectants (Fig. 2j,k). Thus, JAK2 deficiency protected Ma-Mel-54a cells not only from anti-tumour IFNγ activity but also conserved their HLA class I-low phenotype that in turn might have hampered effective T-cell recognition of the tumour cells. Due to a lack of cryopreserved autologous T cells from this patient, the functional significance of the HLA class I-low phenotype could not be investigated in more detail.

**Chromosome 1 alterations predispose to JAK1 deficiency.** Similar to Ma-Mel-54, multiple cell lines were established from distinct melanoma metastases of patient Ma-Mel-61 collected over a period of 2.5 years, allowing us to follow the development of IFNγ resistance in the course of disease. The patient presented in the clinic with stage IV melanoma at the end of 2002 and received IFNα treatment for more than 2 years from December 2002 to January 2005 (Fig. 3a). From three of his metastases, excised under IFNα treatment, the cell lines Ma-Mel-61a, Ma-Mel-61b and Ma-Mel-61c were established. Additional three lesions were resected in 2005 after IFNα therapy, giving rise to the cell lines Ma-Mel-61e, Ma-Mel-61g and Ma-Mel-61h (Fig. 3a). Treatment with IFNγ and subsequent analyses of CD54, HLA class I and PD-L1 surface expression by flow cytometry revealed an upregulation of these markers only for cell lines established before Ma-Mel-61g (Fig. 3b, and Supplementary Fig. 3a).

By targeted sequencing on DNA from Ma-Mel-61g and blood cells as a constitutive normal control we detected a *JAK1* c.1798G>T, JAK1-G600W mutation in 99% of the cells (Fig. 3c, Supplementary Fig. 3b), with the JAK1-G600W exchange affecting a conserved amino acid in the auto-inhibitory pseudokinase domain[17]. None of the IFNγ-sensitive Ma-Mel-61 cell lines showed a *JAK1* mutation (Fig. 3c, Supplementary Fig. 3b). The JAK1-G600W mutant protein was still detectable by western blot (Fig. 3d), but was found to be completely inactive, as no upregulation of pSTAT1, STAT1 and IRF1 was detected in Ma-Mel-61g cells in response to IFNγ treatment (Fig. 3e). Consistently, transfection of Ma-Mel-61g cells with an expression plasmid encoding wild-type JAK1, in contrast to JAK1-G600W, restored IFNγ signalling (Fig. 3f and Supplementary Fig. 3c). Again we detected autologous CD8$^+$ T cells secreting IFNγ in the presence of Ma-Mel-61g cells in the patient's peripheral blood, whose activity might have selected for the outgrowth of IFNγ-resistant cells (Fig. 3g). Accordingly, while proliferation of Ma-Mel-61g cells was not affected by IFNγ, the cells became sensitive upon JAK1 re-expression (Fig. 3h). Analysing the

genetic evolution of JAK1 deficiency, we found a deletion on chromosome 1p, encompassing the region 1p34.3–1p12 (Chr.1:40,061,699-118,932,325) including the *JAK1* gene, to be present in all Ma-Mel-61 cell lines (Fig. 3i). This demonstrated that allelic *JAK1* loss was an early event in the course of disease progression in this patient, predisposing to IFNγ-resistance development, and that a subsequent inactivating *JAK1* point mutation led to complete abrogation of type II IFN signalling in Ma-Mel-61g cells.

**Evolution of T-cell-resistant lesions.** Similar to Ma-Mel-61g, IFNγ signalling was abrogated in Ma-Mel-61h cells due to the homozygous *JAK1* c.1798G>T, JAK1-G600W mutation (Fig. 4a,b). In contrast to Ma-Mel-61g, however, Ma-Mel-61h cells demonstrated a stable HLA class I-negative phenotype. Lack of HLA heavy chains and HLA class I surface expression was observed by western blot and flow cytometry, respectively (Fig. 4b,c). Staining of Ma-Mel-61g and Ma-Mel-61h tissue sections revealed the presence of HLA class I-negative tumour cells in both lesions (Fig. 4d and Supplementary Fig. 3d). Quantification of mRNAs involved in antigen presentation indicated a complete lack in the expression of *HLA-B, HLA-C, TAP1* and *TAP2* in Ma-Mel-61h cells compared with the control cells Ma-Mel-61b (Fig. 4e). Transient JAK1 re-expression and subsequent IFNγ treatment induced *de novo HLA-B, HLA-C, TAP1* and *TAP2* mRNA expression demonstrating a reversible silencing of antigen presentation in Ma-Mel-61h cells (Fig. 4e). This could not be observed for Ma-Mel-61h cells expressing JAK1-G600W, confirming functional inactivity of the mutant protein. Consistently, enhanced expression of specific mRNAs could also be measured for IFNγ-treated Ma-Mel-61g-JAK1 but not for Ma-Mel-61g-JAK1-G600W transfectants (Fig. 4e). As shown in Fig. 4f, the subpopulation of Ma-Mel-61h-JAK1 transfectants demonstrated *de novo* HLA class I surface expression after IFNγ-treatment, resulting in detection of the previously ignored tumour cells by autologous CD8$^+$ T cells (Fig. 4g). Overall, these data demonstrated that JAK1 deficiency in Ma-Mel-61h cells was followed by silencing of antigen presentation, generating a T-cell-resistant melanoma phenotype. Similar results were obtained for JAK2-deficient Ma-Mel-54a cells (Fig. 2g,i,k), suggesting the broader significance of our findings.

**Genetic alterations defined in different melanoma data sets.** The detection of IFNγ-resistant melanoma metastases in our patient cohort led us to assess the presence of alterations in type II IFN signalling pathway components in independent sample

**Figure 2 | JAK2 deficiency blocks HLA class I upregulation by IFNγ.** (**a**) Clinical history of patient Ma-Mel-54. Vertical line, time axis; left, therapeutic regimens; right, primary tumour (PT)/metastases development; arrows indicate cell lines established from metastases Ma-Mel-54a and Ma-Mel-54b; grey box, stage IV disease. (**b**) Mutations defined by targeted sequencing on DNA from melanoma cells and autologous blood cells as wild-type (WT) control (ctrl). Plots of aligned sequencing reads in the location where the *JAK2* c.2876A>C, p.Q959P mutation was identified. WT sequences shown on the bottom, arrows highlight mutation sites. Number of sequencing reads notated on the left; %, frequency of mutations in reads. (**c**) Lysates from Ma-Mel-54a, Ma-Mel-54b and control cells (ctrl) analysed by western blot for JAK2 expression. (**d**) Lysates from IFNγ-treated (48 h) Ma-Mel-54a and Ma-Mel-54b cells analysed for expression of STAT1, pSTAT1, IRF1 and HLA class I heavy chains. (**c,d**) GAPDH, loading control. Representative data from *n* = 2 independent experiments. (**e**) Expression of CD54, HLA class I and PD-L1 on IFNγ-treated (48 h) melanoma cells, measured by flow cytometry. Representative data from *n* = 3 independent experiments. (**f**) SNP results given as allelic distribution of chromosome 9p shown for DNA obtained from Ma-Mel-54a, Ma-Mel-54b and autologous peripheral blood cells as normal control (germline). Loss of one chromosomal allele in region 9p24.3–p13.2 (Chr.9:203,861-37,578,327; hg19) present in both cell lines. Dashed line indicates *JAK2* location at Chr.9p24.1. (**g**) Lysates from IFNγ-treated (48 h) Ma-Mel-54a-JAK2 transfectants analysed by western blot for expression of indicated proteins. (**h**) Real-time cell proliferation in the presence or absence of IFNγ. Bold grey vertical lines indicate addition of IFNγ. (**g,h**) Representative data from *n* = 2 independent experiments. (**i**) Ma-Mel-54a and Ma-Mel-54b cells analysed for specific mRNA expression by quantitative reverse transcription–PCR. (**j**) Ma-Mel-54a-JAK2 transfectants analysed for specific mRNA expression by quantitative reverse transcription–PCR in the presence or absence of IFNγ (48 h). (**i,j**) Relative expression levels given as means ( + s.e.m.) from *n* = 2 independent experiments. (**k**) HLA class I expression on IFNγ-treated (48 h) Ma-Mel-54a cells and Ma-Mel-54a-JAK2 transfectants, measured by flow cytometry. Representative data from *n* = 2 independent experiments.

collections. We screened 287 TCGA (The Cancer Genome Atlas) melanoma tissue samples for mutations in *IFNGR1, IFNGR2, JAK1, JAK2, STAT1* and *IRF1* (ref. 18). In 12.6% (36 of 287) of the samples alterations were identified, including single-nucleotide variations (SNV), small insertions and deletions (Indel) and

homozygous deletions, that in most cases were mutually exclusive (Fig. 5a; Supplementary Data 1, Supplementary Fig. 4). Interestingly, a remarkable fraction of the biopsies (44%, 16 of 36) showed homozygous deletions in the different genes indicating abrogation of type II IFN signalling in tumour cells. Alterations

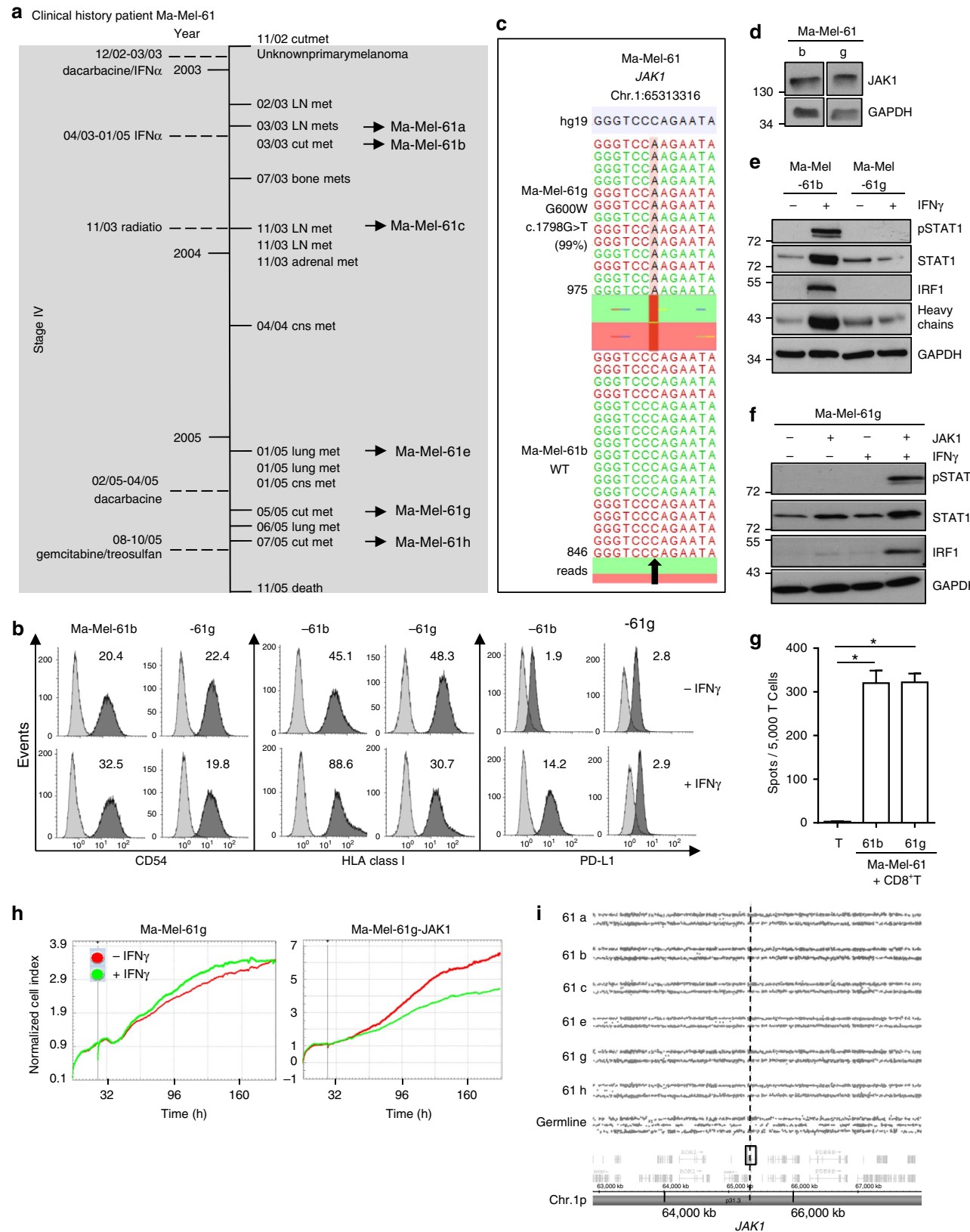

affecting genes of the IFNγ signalling cascade were more frequent in metastatic samples compared with primary tumours (Fig. 5b) and were detectable in metastases from patients receiving neo-adjuvant IFNα and from patients without treatment (Supplementary Data 1). Tumour biopsies with mutations did not show elevated expression of *IFNG* or *CD8A* mRNA in comparison to biopsies without mutations, suggesting a comparable activity of T cells in both types of lesions (Supplementary Fig. 5).

Support of our finding of recurrent genetic alterations in type II IFN signalling pathway genes was also obtained from additional published data sets. A mutation frequency of 22% (11 of 49), based on SNV/Indels and homozygous deletions, was detected in melanoma cell lines studied by the Cancer Cell Line Encyclopedia (CCLE)[19]. Considering only SNV/Indels, the frequency in melanoma tissue samples was 7% (20 of 287) for the TCGA melanoma collection (Fig. 5a)[18], 3% (3 of 91) for melanoma biopsies studied by Krauthammer *et al.*[20] (Fig. 5c), 6% (3 of 49) for melanoma cell lines from CCLE[19] (Fig. 5d) and 10% (12 of 121) for the cell lines analysed by Hodis *et al.*[15] (Fig. 5e). Overall these analyses found mutations to be present in a considerable fraction of melanoma cells.

Our studies in the three melanoma patient models identified allelic *JAK1* and *JAK2* losses as initial genetic alteration predisposing to IFNγ-resistance development. This led us to screen available SNP microarray data from 59 'in-house' melanoma cell lines for loss of heterozygosity (LOH) in the *JAK1* and the *JAK2* locus[21]. As shown in Fig. 5f, LOH for *JAK1* was detected in 25% (15 of 59) and for *JAK2* in 76% (45 of 59) of the cell lines suggesting a high risk of resistance development in the course of an effective anti-tumour T-cell response.

Based on the above sequencing results and functional data, we asked for the impact of alterations in IFNγ signalling genes on the control of disease progression. When assessed in the largest available TCGA melanoma cohort with survival data (479 samples) these alterations were found to have a statistically significant negative impact on patient survival (Fig. 6a). On the other hand, elevated mRNA levels for *STAT1* as well as its downstream target *IRF1*, indicating active IFNγ signalling, were strongly associated with improved overall survival (Fig. 6b). The direct correlation between *IFNG* and *CD8A* mRNA expression pointed to CD8$^+$ T cells as the major cytokine source (Supplementary Fig. 6a,b).

**Mutations emerge before checkpoint blocking therapy.** As resistance to IFNγ signalling could impact therapy efficacy, we screened 59 formalin-fixed, paraffin-embedded (FFPE) tumour samples from patients receiving anti-PD1 therapy for corresponding mutations by targeted sequencing of DNA isolated from macrodissected tumour cells and autologous control blood cells (Table 2). In 19% of the biopsies (11 out of 59) mutations were identified, affecting *IRF1* ($n = 1$), *JAK1* ($n = 5$) and *JAK2* ($n = 5$), some of them clearly inactivating as the stop codon mutation in sample 15-12774 (IRF1 W195*), and the frame shift mutations in specimens D5923-13 (JAK2-D710fs) and 18298-15 (JAK2-T376fs). Furthermore, we determined the mutation frequency in pre-treatment biopsies from patients having received anti-CTLA-4 therapy, in this case evaluating existing exome sequencing data[7]. Mutations in type II IFN signalling pathway genes were identified in 9% of samples (10 out of 110) (Table 2).

Overall, these sequencing results demonstrate that genetic alterations in type II IFN signalling pathway components are present in a considerable number of melanomas. Although we could not detect significant differences in responses to anti-PD1 or anti-CTLA-4 treatment between patients with or without mutations (Supplementary Table 1), it is important to note that two of the three patients with clearly inactivating mutations (15-12774: IRF1 W195*; 18298-15: JAK2 D710fs) showed progressive disease under anti-PD1 treatment, while the third patient (18298-15: JAK2 T376fs) showed a partial response, suggesting that these mutations might have contributed to therapy resistance. Considerably larger cohorts of anti-PD1 and anti-CTLA-4 treated patients will be required to allow conclusive statistical analysis to be performed in future studies.

## Discussion

Remarkable response rates in treatment of metastatic melanoma have been reported for adoptive T-cell transfer[22,23] and therapy with immune checkpoint-blocking antibodies, including anti-PD1 monotherapy[2,3] as well as anti-PD1 and anti-CTLA-4 combination therapy[4,24]. Long-term data from anti-CTLA-4 therapy suggest that a number of patients will show a durable complete response and may even be healed of metastatic disease[25]. Despite the considerable therapeutic potential, not all patients benefit equally well from immunotherapy. Primary as well as acquired therapy resistance is a major concern and the identification of resistance mechanisms is crucial for advancing treatment of melanoma and other malignancies.

The data presented in this work signify that under the selective pressure of an effective T-cell response tumour clones evolve that are considerably less susceptible or even resistant to T-cell effector mechanisms. As such the direct cytotoxic effects of CD8$^+$ T cells mediated by release of cytolytic granules or death receptor engagement are an essential but most likely insufficient part of the overall anti-tumour response that also depends on the secretion of IFNγ. By induction of growth arrest and cell death IFNγ has a broader impact on tumour cells and their

**Figure 3 | Chromosome 1 alterations predispose to JAK1 deficiency.** (**a**) Clinical history of patient Ma-Mel-61. Vertical line, time axis; left, therapeutic regimens; right, metastases development; arrows indicate cell lines established from metastases Ma-Mel-61a, Ma-Mel-61b, Ma-Mel-61c, Ma-Mel-61e, Ma-Mel-61g and Ma-Mel-61h; grey box, stage IV disease. (**b**) Surface expression of CD54, HLA class I, and PD-L1 on IFNγ-treated (48 h) Ma-Mel-61b and Ma-Mel-61g cells, measured by flow cytometry. Representative data from $n = 3$ independent experiments. (**c**) Mutation defined by targeted sequencing on DNA from Ma-Mel-61g cells. Plots of aligned sequencing reads in the location where the *JAK1* c.1798G > T, p.G600W mutation was identified. Arrow highlights mutation site in Ma-Mel-61g or corresponding wild-type site (WT) in Ma-Mel-61b cells. Number of sequencing reads notated on the left; %, frequency of mutation in reads. (**d**) Lysates from Ma-Mel-61b and Ma-Mel-61g cells analysed by western blot for JAK1 expression; GAPDH, loading control. Representative data from $n = 3$ independent experiments. (**e**) Lysates from IFNγ-treated (48 h) melanoma cells analysed by western blot for expression of STAT1, pSTAT1, IRF1 and HLA class I heavy chains; GAPDH, loading control. Representative data from $n = 3$ independent experiments. (**f**) Lysates from IFNγ-treated (48 h) *JAK1*-transfected Ma-Mel-61g cells analysed for expression of the indicated proteins. Representative data from $n = 3$ independent experiments. (**g**) IFNγ release by autologous CD8$^+$ T cells in the presence of melanoma cells, measured by ELISpot assay. Means and s.e.m. (error bars) from $n = 4$ independent measurement. Statistical significant differences defined by paired Student's *t*-test are indicated, *$P < 0.05$. (**h**) Ma-Mel-61g and Ma-Mel-61g-JAK1 cells subjected to impedance-based real-time measurement of proliferation in the presence or absence of IFNγ. Addition of IFNγ indicated by bold grey vertical lines. Representative data from $n = 3$ independent experiments. (**i**) SNP results given as allelic distribution of chromosome 1p shown for DNA obtained from the different Ma-Mel-61 cell lines and autologous blood cells as normal control (germline). Loss of one chromosomal allele in the region 1p34.3–1p12 (Chr.1:40,061,699-118,932,325; hg19) present in all cell lines. Dashed line indicates location of *JAK1* at Chr.1p31.3.

microenvironment[9–11,13,14]. Accordingly, analyses on TCGA melanomas revealed a strong association between patient survival and elevated mRNA levels of *STAT1* and its downstream target *IRF1*, indicating IFNγ-dependent pathway activation. Furthermore, the strong correlation between *IFNG*

and *CD8A* mRNA in melanomas, both known as favourable prognostic markers[26–29], argues for CD8[+] T cells as a major IFNγ source. However, a significant contribution by other lymphocytes such as CD4[+] T cells of the Th1 phenotype or natural killer cells cannot be excluded[13,30].

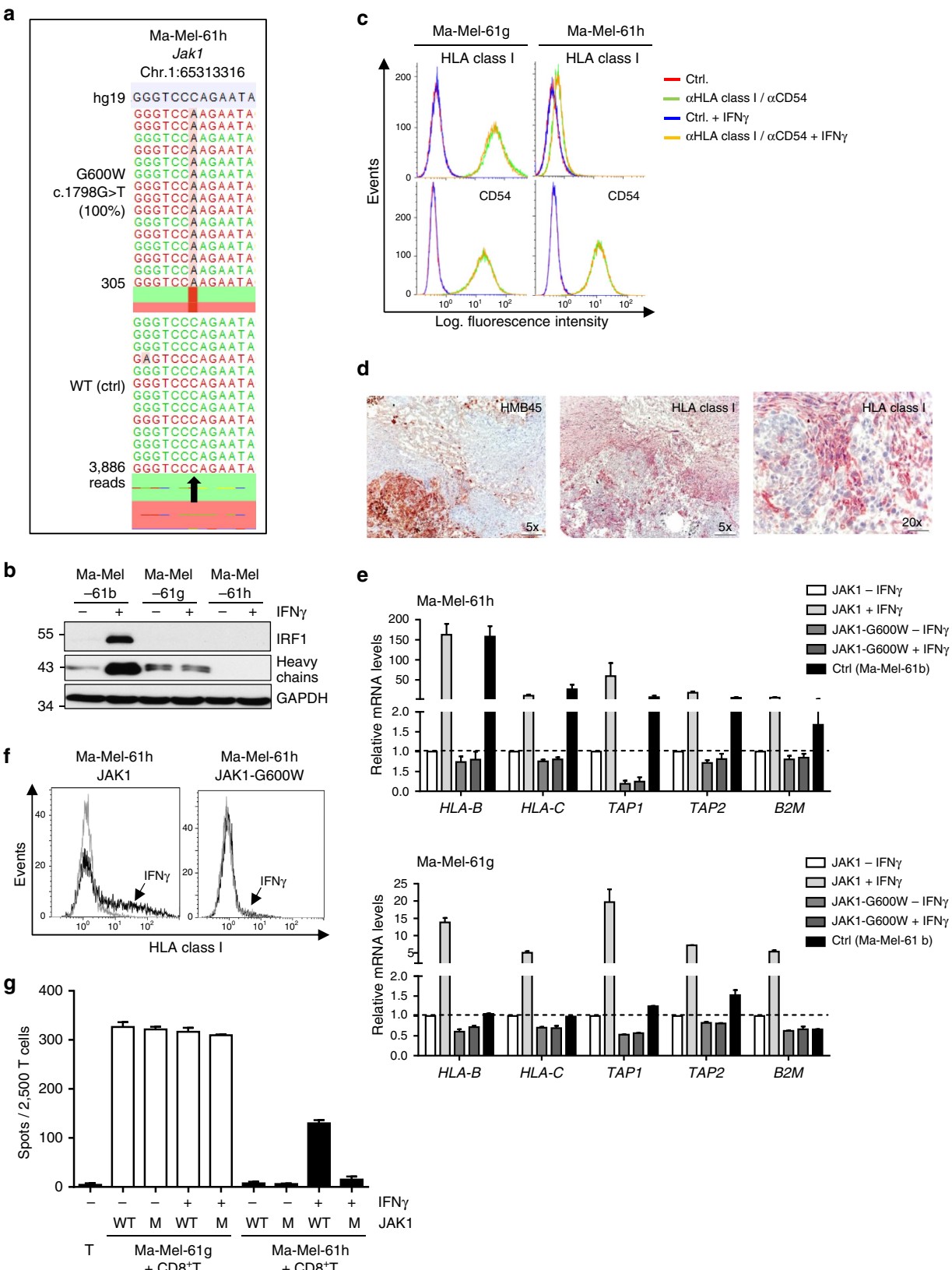

Assuming that IFNγ exerts a strong selective pressure on tumour cells we screened an 'in-house' collection of short-term cultured melanoma cell lines for mutations in genes of the IFNγ signalling pathway and detected *JAK1, JAK2* and *STAT1* alterations in cells and corresponding tumour tissue from 6 out of 47 patients. In two heterozygous *JAK1* mutants IFNγ signalling was still active but was strongly impaired and no longer detectable in homozygous STAT1 and JAK1/2 mutants, respectively. JAK1/2-deficient tumour cells emerged in disease stage IV metastases under/after immunotherapy. The different treatments, including IFNα, IL2 and combinations thereof, might have induced or boosted the effector functions of tumour-reactive CD8[+] T cells, favouring mutant outgrowth. Indeed, we demonstrated that JAK1/2 loss protected melanoma cells from anti-proliferative and pro-apoptotic IFNγ activity. Since JAK1 is a component also of the type I IFN signalling pathway, an additional selective pressure of IFNα on tumour cells cannot be excluded[8,31]. In contrast to the JAK1/2-deficient melanoma cells, *STAT1* mutant cells were established from a treatment-naive stage III lymph node metastasis suggesting that in this case spontaneous anti-tumour T-cell responses enriched these cells. In addition to melanoma, inactivating mutations in genes related to IFNγ signalling, in particular *JAK1*, have recently been described for microsatellite instable endometrial and colorectal cancers, arguing for a contribution to disease progression also in other malignancies[32–37].

In our patient models, JAK1 deficiency originated from an initial chromosome 1p aberration causing mono-allelic *JAK1* loss in melanoma cells and a subsequent mutation inactivating the remaining *JAK1* allele. Losses of the short arm of chromosome 1 are not uncommon in cutaneous and uveal melanoma with larger deletions occurring in around 10% of cutaneous melanomas[38,39]. More focal deletions as well as copy number neutral losses of heterozygosity may occur, that we detected at a high frequency in a cohort of 59 melanoma cell lines[21], showing also a very high frequency of allelic *JAK2* losses (76%). All of these alterations in addition to gene mutations would predispose to *JAK1/JAK2* inactivation and IFNγ-resistance in tumours if put under selective pressure by the immune system.

Interestingly, our data demonstrate that IFNγ-resistant JAK1/2-deficient melanoma cells progress to a 'higher level' of immunotherapy resistance. We provide evidence for the first time that IFNγ-resistant HLA class I-positive metastases can evolve into HLA class I-negative lesions thereby gaining complete CD8[+] T-cell resistance. The HLA class I-negative phenotype is caused by a coordinated silencing of genes involved in antigen presentation (*HLA-B, HLA-C, TAP1, TAP2, B2M*). Downregulation of this set of genes has previously been reported for melanoma and other tumour entities. The underlying molecular silencing mechanisms remain unclear but are most likely of epigenetic nature[40–42]. IFNγ is well known for its role in upregulating antigen processing and presentation thereby augmenting the detection and elimination of malignant cells by tumour antigen-specific CD8[+] T cells[43,44]. However, in case of JAK1/2 deficiency IFNγ-induced restoration of antigen presentation in tumour cells is abrogated. Phenotypically HLA class I-negative JAK1/2-deficient metastases share features with tumours lacking HLA class I surface expression due to inactivating *B2M* mutations as described by us and others[33,45–48]. HLA class I-negative metastases will be resistant towards any type of immunotherapy that is dependent on the activity of HLA class I-restricted tumour antigen-specific CD8[+] T cells, including adoptive cell therapy and checkpoint modulators. However, in contrast to *B2M* mutants, melanoma cells of the regulatory HLA class I-negative phenotype can regain HLA class I expression to adapt to specific environmental conditions such as metastatic sites (for example, lung, liver) enriched for natural killer cells that are specialized in recognition and killing of HLA class I-negative malignant cells[49].

Recently, resistance to anti-PD1 and anti-CTLA-4 therapy has been associated with sustained IFNγ signalling upregulating ligands for multiple inhibitory receptors on T cells, as well as IFNγ resistance protecting from cytokine-induced cell cycle arrest/apoptosis[33–35,50]. It will be of importance to determine how far the alterations we detected in pretreatment biopsies will undergo positive selection in tumours recurring upon anti-PD1 therapy or whether mutations will evolve *de novo* as recently described[33]. Of equal importance will be the identification of novel IFNγ-resistance mechanisms. Epigenetic factors as well as altered expression of negative IFNγ pathway regulators in tumour cells or microenvironmental influences could have an additional relevant role in conferring resistance or reduced sensitivity to IFNγ[51,52]. Furthermore, the resistance mechanisms could go beyond IFNγ and apply to other cytokines such as TNFα[13,14]. In this regards, the combined action of IFNγ and TNFα has been demonstrated to destroy tumour cells and their stroma thereby essentially contributing to the eradication of established mouse tumours[53].

It will be a considerable future challenge to identify all mutations associated with IFNγ resistance and to define the coevolution of HLA class I expression in longitudinal melanoma biopsies. Only this strategy will ensure that patients receive the most promising treatment options and be switched to other therapeutic regimens if IFNγ resistance develops, for example, oncogenic pathway inhibitors that could eliminate resistant tumour clones and allow the patients to reinitiate immunotherapy.

**Figure 4 | IFNγ-resistant melanoma evolves into a T-cell-resistant lesion. (a)** Mutation defined by targeted sequencing on DNA from Ma-Mel-61h cells and autologous blood cells as wild-type (WT) control (ctrl). Plots of aligned sequencing reads in the location where the *JAK1* c.1798G > T, p.G600W mutation was identified. WT sequence shown on the bottom, arrow highlights mutation or corresponding wild-type (WT) site. Number of sequencing reads notated on the left; %, frequency of mutations in reads. **(b)** Lysates from IFNγ-treated (48 h) Ma-Mel-61b, Ma-Mel-61g and Ma-Mel-61h cells analysed by western blot for expression of IRF1 and HLA class I heavy chains; GAPDH, loading control. Representative data from n = 3 independent experiments. **(c)** HLA class I and CD54 surface expression on IFNγ-treated (48 h) Ma-Mel-61g and Ma-Mel-61h cells, measured by flow cytometry. Representative data from n = 3 independent experiments. **(d)** Immunohistochemical staining of serial cryostat tissue sections from metastasis Ma-Mel-61g for melanoma marker HMB45 and HLA class I. **(e)** Ma-Mel-61h and Ma-Mel-61g cells, transfected with expression plasmids encoding wild-type JAK1 or mutant JAK1-G600W, analysed for specific mRNA expression by quantitative reverse transcription–PCR in the presence of absence or IFNγ (48 h). Ma-Mel-61b cells served as a control (ctrl). Relative expression levels given as means ( + s.e.m.) from n = 2 independent experiments. **(f)** HLA class I surface expression on IFNγ-treated (48 h) Ma-Mel-61h-JAK1 and Ma-Mel-61h-JAK1-G600W transfectants, measured by flow cytometry. Representative data from n = 2 independent experiments. **(g)** Ma-Mel-61h and Ma-Mel-61g cells, transfected with expression plasmids encoding wild-type JAK1 (WT) or mutant JAK1-G600W (M), analysed for recognition by autologous CD8[+] T cells in the presence or absence of IFNγ (48 h). T-cell activation measured as IFNγ release by ELISpot assay. Representative data from n = 2 independent experiments.

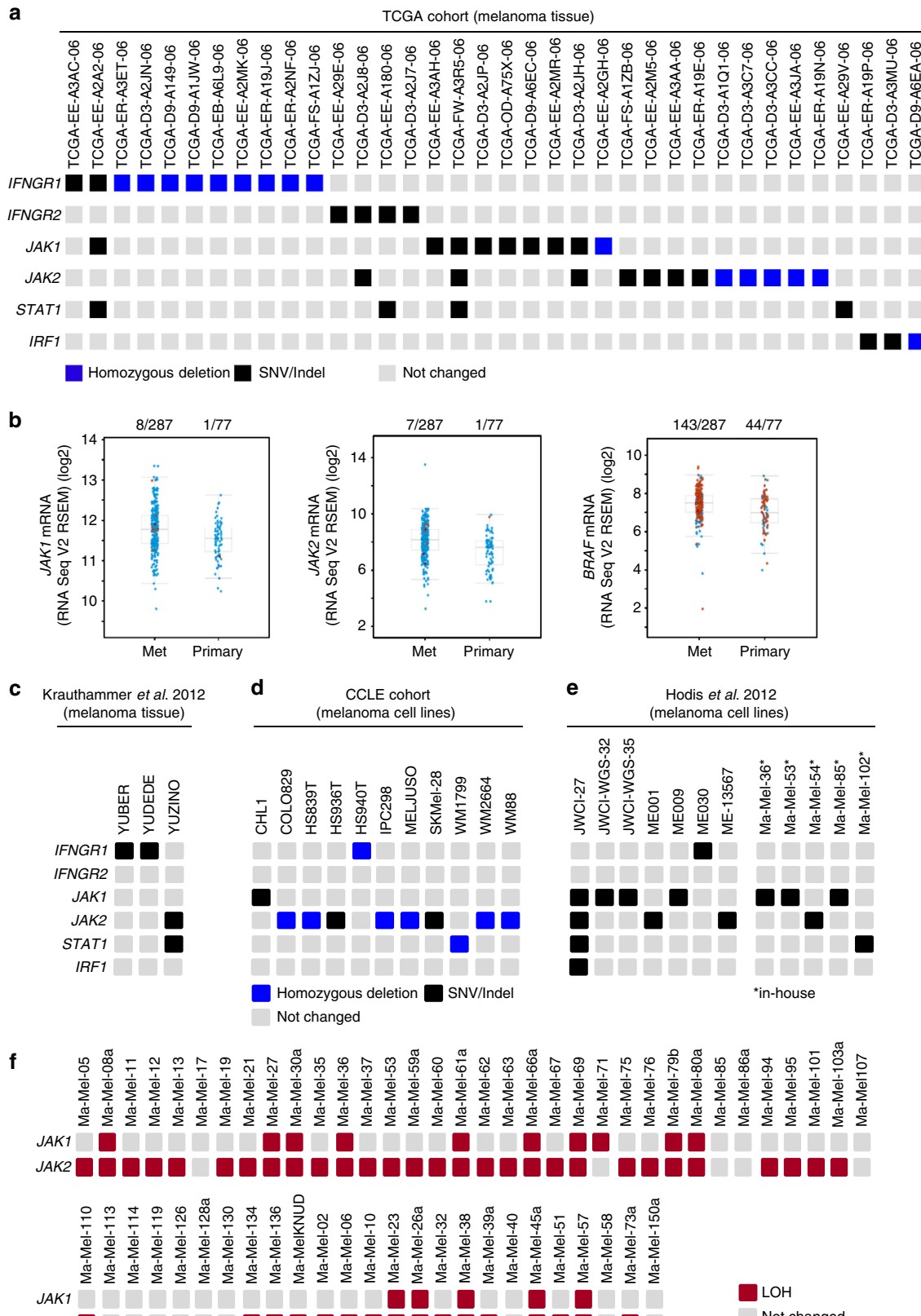

**Figure 5 | Mutations in melanoma tissue samples and cell lines.** (**a**) Genetic alterations (homozygous deletions, single-nucleotide variants [SNV], small insertions and deletions [Indel]) in components of the type II IFN signalling pathway in TCGA melanoma samples ($n = 287$, accessed 31/12/2016). (**b**) Mutations in *JAK1, JAK2* and *BRAF* in the TCGA melanoma dataset stratified by tissue origin (primary tumours vs metastases). *y* axis, indicates expression of corresponding mRNA. Each circle represents a sample, samples harbouring SNV or Indels are shown in red ($n = 287$, accessed 30/12/2016). (**c–e**) Recurrent mutations of type II IFN signalling pathway genes in published datasets[15,19,20]. (**f**) SNP array data from 59 melanoma cell lines[21] show LOH at the *JAK1* (Chr. 1p31.3) and *JAK2* (Chr. 9p24.1) locus.

## Methods

**Patients samples.** Peripheral blood samples and tumour tissues were collected after written informed patient consent with institutional review board approval. Melanoma cell lines were established from excised metastatic lesions. Cell lines were confirmed to be mycoplasma-free in monthly intervals and authenticated by genetic profiling on genomic DNA at the Institute for Forensic Medicine

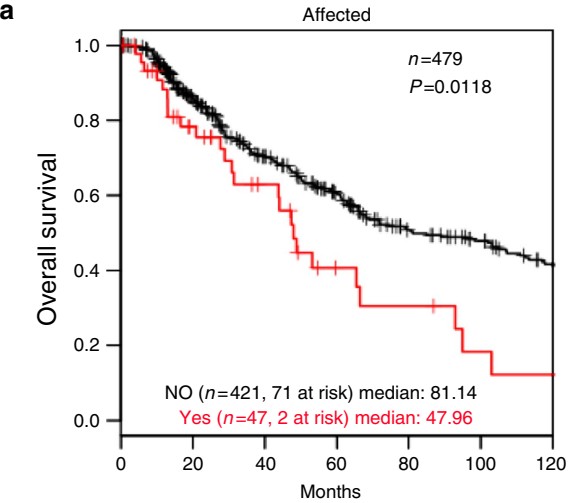

(University Hospital Essen) using the AmpFLSTR-Profiler Plus kit (Applied Biosystems). Melanoma cells were cultured in RPMI1640 or DMEM medium with L-glutamine (Gibco/Life technologies) and 10% fetal calf serum. Cells were seeded and rested overnight followed by addition of IFNγ (500 U ml$^{-1}$, Boehringer Ingelheim) or IFNα2b (1,000 U ml$^{-1}$, Essex Pharma) and incubation for indicated periods.

**Immunohistochemistry.** Serial cryostat tissue sections were stained with antibodies specific for HLA-DR,-DP,-DQ, kindly provided by S. Ferrone[54], HMB-45 (Dako), HLA class I antigen complexes (W6/32; Dianova) in combination with a Polymer Kit containing an AP-coupled secondary antibody (ZytoChem-Plus AP Polymer Kit, Zytomed).

**Isolation of genomic DNA.** Five to ten 10 μm-thick sections of FFPE tissue were deparaffinized according to the following protocol: 2 steps of 10 min xylene, 5 min 100% ethanol, 5 min 95% ethanol, 5 min 70% ethanol, rinsing in water. After drying, tumour tissue was manually macrodissected from the sections. Genomic DNA from tissue as well as 200 μl of whole blood (normal control) was isolated using the QIAamp DNA Mini Kit (Qiagen, Hilden, Germany) according to the manufacturer's instructions. The same kit was used for DNA isolation from pelleted cultured tumour cells, Epstein-Barr virus-transformed B lymphocytes and peripheral blood mononuclear cells.

**Targeted sequencing.** A custom amplicon-based sequencing panel covering 12 genes of the interferon pathway (Supplementary Table 2) was designed and prepared applying the GeneRead Library Prep Kit from QIAGEN according to the manufacturer's instructions. Individual samples were barcoded using a kit from New England Biosciences and 24 samples sequenced in parallel on an Illumina MiSeq Next Generation Sequencer. Sequencing analysis was performed applying the CLC Cancer Research Workbench from QIAGEN. After trimming the primer sequences, the sequence reads were aligned to the human genome assembly 19 (hg19). Analysis for both in/dels and SNVs followed. SNPs were filtered out by cross-referencing the dbSNP database, the 1,000 genomes database and in individual cases manually. Mutations affecting the coding region of the gene were considered if predicted to result in non-synonymous amino acid changes, overall coverage of the mutation site was ≥30 reads, >5 reads reported the mutation variant and the frequency of mutated to unmutated reads was ≥10%.

**SNP array analysis.** SNP arrays were performed using the CytoScan HD Array from Affymetrix. Hybridization was done according to the manufacturer's protocol and data analysis performed applying the program Chromosome Analysis Suite from Affymetrix.

**Quantitative real-time PCR.** Total mRNA was isolated from tumour cells using the RNeasy plus Mini Kit (Qiagen), in combination with RNase-free DNase Set (Qiagen) according to the manufacturer's instructions. Reverse transcription, TaqMan-based real-time PCR and calculation of relative expression were performed as described previously[55]. Taqman assay systems specific for HLA-B, HLA-C, TAP1, TAP2, B2M, GAPDH were purchased from Thermo Fisher. In all experiments the amount of specific mRNA was normalized to endogenous GAPDH mRNA levels.

**Analyses of published melanoma data sets.** Kaplan–Meier survival plots, log-rank tests and multivariate Cox-regressions based on differentially expressed genes from TCGA melanoma samples were assessed using the UZH cancer browser samples[56], using the 10% percentile for comparing samples with the lowest and the highest mRNA expression for IFNG, STAT1, IRF1 and CD8A. Information on mutations, including single-nucleotide variants (SNV), small insertions/deletions (Indels) and copy number variants (considering only homozygous deletions) in genes of the type II IFN signalling pathway (IFNGR1, IFNGR2, JAK1, JAK2, STAT1, IRF1) were obtained from TCGA skin cutaneous melanoma (SKCM) samples with complete mutation data (n = 287, accessed 31/12/2016). Survival analyses based on mutations and aberrant protein expression were based on the extended SKCM TCGA data set (n = 487) and clinical data obtained from cBioPortal, generating

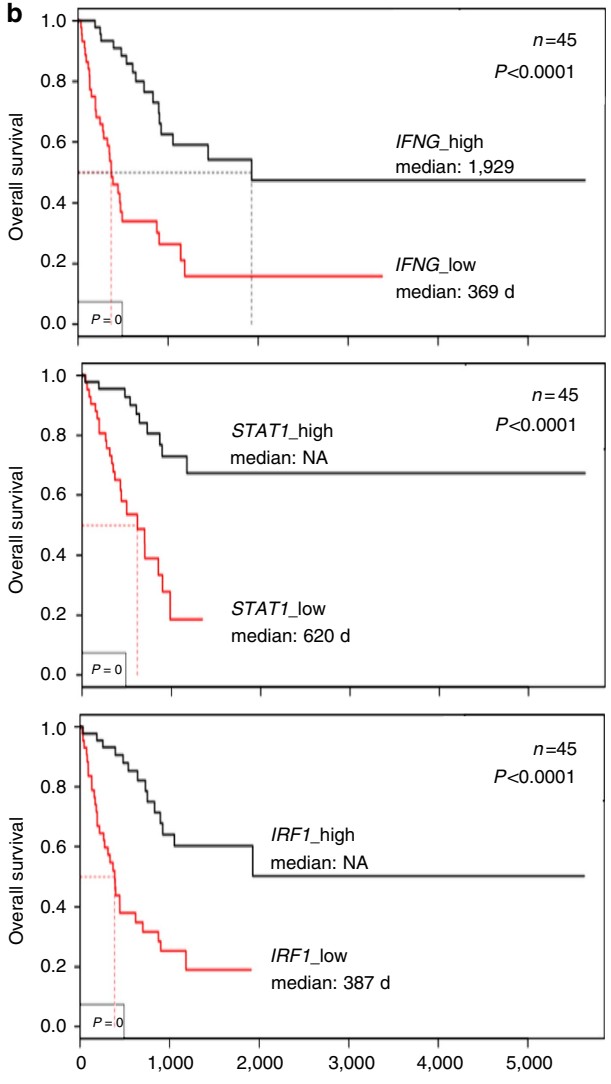

**Figure 6 | Alterations in IFNγ pathway signalling impact on patient survival.** (**a**) Truncating mutations, homozygous deletions and low protein levels of IFNGR1, IFNGR2, JAK1, JAK2, STAT1 and IRF1 define a subset of melanoma patients in the TCGA data set with decreased survival. Of 479 patients with data on aberrations, 468 had available survival data. Log-rank P value shown. (**b**) Kaplan–Meier survival curves for IFNG, STAT1 and IRF1 expressing TCGA melanomas[56], P values shown from Log-rank tests. Multivariate Pvalues corrected for age: IFNG P = 1.79e-05, n = 90, 47 events; STAT1 P = 3.82e-05, n = 90, 33 events; IRF1 P = 2.52e-05, n = 90, 44 events.

**Table 2 | Mutations in biopsies from patients receiving anti-PD1 (n = 59) and anti-CTLA4 (n = 110) immunotherapy.**

| Material | Sample | Gene | AA change | cDNA change | Treatment | Response |
|---|---|---|---|---|---|---|
| Patient | 15-12774 | IRF1 | W195* | 584G > A | Nivo | PD |
| Patient | D5923-13 | JAK2 | D710fs | 2123_2124insC | Pemb | PD |
| Patient | 31968-14 | JAK1 | R681Q | 2042G > A | Pemb | PD |
| Patient | 14-28049 | JAK1 | E501K | 1501G > A | Nivo | CR |
| Patient | D788-15 | JAK1 | E903K | 2707G > A | Nivo | PD |
| Patient | 13-12191 | JAK2 | H103Y | 307C > T | Nivo | PR |
| Patient | 32616-13 | JAK2 | S15F | 44C > T | Nivo | PR |
| Patient | 18298-15 | JAK2 | T376fs | 1125delA | Pemb | PR |
| Patient | 5870-14† | JAK2 | P121S | 361C > T | Nivo | SD |
| Patient | D2788-15 | JAK1 | K249N | 747G > T | Nivo | Unknown |
| Patient‡ | 343-16§ | JAK1 | L149F, T147I | 445C > T, 440C > T | Nivo | PD |
| Patient | 132 | IFNGR1 | D465N | 1393G > A | Ipilimumab | PR |
| Patient | 4 | JAK1 | R110_splice | e5-1 | Ipilimumab | PR |
| Patient | 14 | JAK1 | G590R | 1768G > C | Ipilimumab | PD |
| Patient | 151 | JAK1 | V715M | 2143G > A | Ipilimumab | PD |
| Patient | 151 | JAK1 | G655D | 1964G > A | Ipilimumab | PD |
| Patient | 37 | JAK1 | G182E | 545G > A | Ipilimumab | PD |
| Patient | 4 | JAK2 | R138Q | 413G > A | Ipilimumab | PR |
| Patient | 163 | JAK2 | G164A | 491G > C | Ipilimumab | PD |
| Patient | 110 | IRF1 | R314W | 940C > T | Ipilimumab | PD |
| Patient | 82 | IRF1 | S221F | 662C > T | Ipilimumab | PD |

AA, amino acid; cDNA, complementary DNA; CR, complete response; fs, frameshift; ins, insertion; Nivo, Nivolumab; PD, progressive disease; Pemb, pembrolizumab; PR, partial response; SD, stable disease; splice, splice site mutation.
*Nonsense—stop codon mutation.
†Additionally a IFNGR1, E195K, 685G > A mutation.
‡Relapse under treatment.
§Additionally a IFNGR1, G129E, 461G > A and IFNGR2, T187I, 503C > T mutation.

Kaplan–Meier survival plots and log-rank tests in R (R Development Core Team; http://www.R-project.org). Coexpression plots were obtained for TCGA SKCM samples using cBioPortal[57]. Mutation calls, including SNV and Indels, from 110 patients before anti-CTLA-4 antibody therapy were reported previously[7]. Additional mutation data for melanoma tissues and cell lines with SNV/Indels[15,20] as well as for melanoma cell lines including also homozygous deletions, as part of the CCLE project[19] were assessed using cBioPortal[57].

SNP array data from 59 melanoma cell lines analysed with the 250 k StyI SNP array of the Affymetrix GeneChipV 500 K array set (Affymetrix, Santa Clara, CA), GEO accession number GSE17534 (ref. 21) were assessed for LOH using Affymetrix genotyping console software. In 44 cases, SNP data from corresponding germline DNA were available for paired analysis. In the remaining cases, the SNP data from tumour samples were compared with baseline values obtained from combined analysis of the SNP data from the 44 available germline cases.

**Western blot.** Proteins from tumour cell lysates were separated by SDS–polyacrylamide gel electrophoresis, blotted on nitrocellulose membranes and probed with the following primary antibodies: anti-STAT1 (Santa Cruz, clone M-22, 1:1000) and anti-pSTAT1 (Cell Signaling, clone 58D6, 1:1,000), anti-IRF1 (Santa Cruz, clone H-205, 1:500) and anti-GAPDH (Cell Signaling, 14C10, 1:5,000). HC10 (1:1,000) was used for detection of β2m-free HLA heavy chains[58,59]. After washing, membranes were incubated with the appropriate secondary antibodies linked to horseradish peroxidase. Antibody binding was visualized with the enhanced chemiluminescence (ECL) system. Full scans of western blots are depicted in Supplementary Figs 7–9.

**Flow cytometry.** The following directly labelled antibodies were used for staining of cellular surface markers: anti-HLA-ABC-APC (eBiosciences, clone W6/32; 1 μl), anti-CD54-PE (Beckmann Coulter, clone 84H10; 2.5 μl), anti-PD-L1-PE (Biolegend, clone 29E2A3; 5 μl) and anti-HLA-DR-PC7 (Beckmann Coulter, clone Immun-357; 2.5 μl). After fixation, stained cells were analysed by flow cytometry on a Gallios flow cytometer (Beckmann Coulter) and Kaluza (Beckman Coulter) software, respectively, for data analysis. In order to isolate specific Ma-Mel-36 subpopulations, cells were stained with anti-HLA-DR-PC7 and sorted based on the specific expression of the surface markers by flow cytometry on an Aria II cell sorter and the FACS Diva software (BD Biosciences).

**Plasmid generation and transfection.** Wild-type JAK1 was amplified using Phusion High-Fidelity DNA Polymerase (NEB) and the following primers: JAK1-SPAfo: 5′-ATCGTCCTCGAGATGCAGTATCTAAATATAAAA-3′ and JAK1-SPAre: 5′-ATTGCTCATATGTTTTAAAAGTGCTTCAAATCC-3′. After restriction digest with XhoI and NdeI, the PCR product was ligated into the PMZ3F vector (kindly provided by the laboratory of Jack Greenblatt, University of

Toronto)[60]. Protein expression was verified by immunoblotting using a Flag-specific antibody (Sigma). The point mutation was introduced using Quikchange mutagenesis (Agilent) according to the manufacturer's protocol and the following primers: JAK1G600Wfo: 5′-ACACACATCTATTCTTGGACCCTGA TGGATTA-3′ and JAK1G600Wre: 5′-TAATCCATCAGGGTCCAAGAATAGAT GTGTGT-3′. The intended point mutations were verified by DNA sequencing and protein expression was examined by immune blotting. Lipofectamine (Life Technologies) was used for plasmid transfection of melanoma cells. After 48 h, cells were harvested and subjected to further analyses or treated with G418 for enrichment of transfectants.

**Real-time proliferation assay (xCELLigence).** For background measurement 50 μl medium was added to an E-Plate 96 (Roche). Subsequently, melanoma cells were seeded in an additional volume of 100 μl medium. Cell attachment was monitored using the RTCA SP (Roche) instrument and the RTCA software Version 1.1 (Roche). After 20–24 h cells were treated with IFNγ (500 U ml $^{-1}$) or left untreated, followed by incubation for 7 d at 37 °C. All experiments were performed in duplicates. Changes in electrical impedance were expressed as a dimensionless cell index value, which derives from relative impedance changes corresponding to cellular coverage of the electrode sensors, normalized to baseline impedance values with medium only.

**Expansion of autologous tumour-reactive T cells.** Tumour-reactive T cells were expanded following a previously described protocol[47]. Briefly, CD8 $^+$ T lymphocytes were isolated from cryopreserved peripheral blood mononuclear cells using anti-CD8 MicroBeads (Miltenyi Biotech). Isolated T cells (1 × 10⁶) were co-cultured in 24-well culture plates with 1 × 10⁵ irradiated (100 Gy or 120 Gy) autologous tumour cells per well in 2 ml of AIM-V (GIBCO/BRL) supplemented with 10% (vol/vol) human AB serum. Medium was supplemented with IL-2 (250 U ml $^{-1}$) on day 3. CD8 $^+$ T cells were restimulated at weekly intervals with irradiated melanoma cells. After two rounds of restimulation, T cells were subjected to ELISpot assays.

**IFNγ ELISpot assay.** IFNγ enzyme-linked immunospot (ELISpot) assay was performed as previously described[61]. Briefly, multiscreen-HA plates (Millipore, Bedford, MA) were coated with 5 μg ml $^{-1}$ anti-hIFNγ-mAb 1-D1K (Mabtech). T cells were seeded in RPMI medium and added at indicated numbers to 1 × 10⁴ tumour cells per well. After 20–24 h incubation at 37 °C in 5% CO₂, a biotinylated secondary anti-hIFNγ antibody (1 μg ml $^{-1}$, clone 7-B6-1, Mabtech) was added and spots were developed by sequential addition of 1:1,000 diluted ExtrAvidin alkaline phosphatase and BCIP/NBT Liquid Substrate System (Sigma-Aldrich). Spot numbers were determined with the AID EliSpot reader (AID Diagnostika).

**Data availability.** SNP array data files are accessible at the NCBI GEO database. Cell lines used in this study can be obtained by MTA.

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

## Acknowledgements

Funding: This study was supported by the Deutsche Krebshilfe to A.P. and D.S. ('Translational Oncology' 111546) and to Su.H. (70111402); M.T. and A.M.W. receive funding from the DFG through TRR60 and GRK1949. The results shown here are in part based on data generated by the TCGA Research Network: http://cancergenome.nih.gov/.

## Author contributions

A.S. established patient models, performed targeted sequencing, analysed data; F.Z. carried out T-cell assays; C.H. performed T-cell assays and IHC; N.P. performed western blots, PCR, apoptosis and proliferation assays; R.M. carried out western blots, transfections, proliferation assays; N.S. supported targeted sequencing; B.R. helped with western blots; N.B. carried out IHC; Se.H. generated expression plasmids; B.W., R.G., J.U., C.L. and H.G. provided patient samples; L.K.-H. and M.Z. performed SNP arrays and data analyses; A.M.W. carried out cell sorting; M.T. supported plasmid constructions, edited the manuscript; Su.H. performed bioinformatics; B.S. collected patient samples, edited the manuscript; D.S. established patient models, contributed to conception and design of the study; K.G. contributed to conception and design of the study, carried out data analyses, edited manuscript; A.P. designed experiments, analysed data, wrote manuscript.

## Additional Information

**Accession code:** SNP array data files are accessible at the NCBI GEO database under GSE96884.

**Competing interests:** B.W.: grants from Merck/MSD, Bristol Myers Squibb, personal fees from Philogen, Roche. J.U.: advisory boards and on speakers bureaus of Amgen, BMS, GSK, MSD, Novartis, and Roche. R.G.: research support from Novartis, Pfizer, Johnson&Johnson; honoraria for lectures from Roche Pharma, Bristol-MyersSquibb, GlaxoSmithKline, Novartis, Merck Serono, MSD, Almirall-Hermal, Amgen, Galderma, Janssen, Boehringer Ingelheim; paid advisory role for Roche Pharma, Bristol-MyersSquibb, GlaxoSmithKline, Novartis, MSD, Almirall-Hermal, LEO, Amgen, Pfizer. C.L.: Advisory/Consultant or speakers fee from BMS, Roche, Amgen, Novartis, Pierre Fabre, Biontech, MSD. B.S.: advisory board or honoraria from Novartis, Roche, Bristol-Myers Squibb and MSD Sharp & Dohme, research funding from Bristol-Myers Squibb and MSD Sharp & Dohme, travel support from Novartis, Roche, Bristol-Myers Squibb and AMGEN. D.S.: advisory board or honoraria from Roche, Genentech, Novartis, Amgen, GlaxoSmithKline, Bristol-Myers Squibb, Boehringer Ingelheim, and Merck Sharp & Dohme. All other authors declare no competing financial interests.

