## [Peer Review File · Nature Communications]

Reviewers' comments:

Reviewer #1 (Remarks to the Author):

A number of patients do not benefit from immunotherapy despite its considerable potential, In the paper entitled " Acquired IFN resistance impairs anti-tumor immunity in melanoma", Antje Sucker and colleagues discuss the necessity of identifying resistance mechanisms for advancing treatment of melanoma. The authors demonstrate the evolution of IFN resistance in melanoma cells. They tested whether melanoma cells from patients responding to immunotherapy are sensitive to the anti-proliferative and pro-apoptotic effects of IFN and that continuous cytokine exposure selects for the outgrowth of IFN-resistant tumor subclones. Their results show that various treatments meant to increase CD8+ T cell anti-tumor activity, such as checkpoint blockade inhibitors (PD-1, CTLA4), mediate the positive selection of tumor subclones which are immune to CD8+ anti-tumor activity by resistance to interferon- γ signaling. Interferon- γ promotes apoptosis and arrest cell growth in tumor cells with intact Interferon- γ signaling, whereas tumor cells with mutations in interferon associated downstream genes, such as kinases JAK1 and JAK2 and the transcription factor STAT1, do not undergo these effects; in addition, tumor cells bearing mutations for these genes do not upregulate surface HLA class I and PD-L1 as a result of Interferon- γ stimulation. The interferon-resistance cells are positively selected by the enhanced CD8+ T cell activity as a result of the treatment, and may partially explain the relapse seen in patients undergoing immunotherapy.

Although the findings and conclusions are important and would interest the melanoma research community, the data shown in the paper is mostly not novel. Many of the findings, including the emergence of JAK mutated subclones following treatment and their resistance to Interferon- γ effects on cell growth, were already reported previously by Zaretski et al, NEJM 2016. Similar findings were also published recently by Gao et al, Cell 2016, which reported even further mutations in other interferon related genes. While the authors slightly expand the principle of Interferon- γ resistance to other genes in the interferon cascade, they bring little new concepts regarding the resistance mechanisms. Therefore the manuscript in its current form is not suitable for a publication in Nature Communications.

Major comments:

1. Figure 1- the title of the figure suggests the outgrowth of interferon-resistance cells following anti-PD1 treatment. However in this figure the only indication of the resistant cells is the minor populations that did not upregulate CD54 and PDL1 (35% and 28% respectively). The authors did not check if these minor population phosphorylate STAT1 or undergo apoptosis as performed in figure 1E-F for the whole population, or as they did for the HLA-DR+ and HLA-DR- cells in figure 6. It is recommended that figure 1G should go to supplementary data, since it is not fully indicate Interferon- γ resistance.
2. Figure 1 and Figure 2 are very similar. Both show responsiveness of tumor cell lines derived before and following treatment with checkpoint inhibitors, each from one individual patient. It is not clear what new data does figure 2 add on top of figure 1, other than different treatment course due to the different clinical history of the two patients.
3. Figure 3, showing favorable clinical outcomes of the presence of CD8+ T cells and poor outcomes in their absence, is reminiscent of previous data in the literature (Taylor et al, Journal of clinical oncology 2007; Azimi et al, Journal of clinical oncology 2012, to name a few). Therefore it is recommended to cite these publications and move the figure to the supplementary data.
4. In figure 4, the authors show that Ma-Mel-102 cells are not responding properly to interferon, however they do not show ablation of the pro-apoptotic/ anti-proliferative effect of interferon. Thus, a functional readout for the STAT1 mutation is lacking. Similarly, a functional readout is missing in figure 5.
5. In figure 5, Ma-Mel-54a cells are HLA class I low, however Ma-Mel-54b do express HLA class I levels which are comparable with other cell lines described in the paper. The two cell lines have the exact JAK2 mutation (Q959P), however the difference in HLA class I expression is not

explained. This is an important point since the authors claim that the JAK2 mutation keep HLA class I levels low. Figure 5D suggest that low HLA class I levels can be restored in a manner that is independent of interferon- γ , this should at least be discussed.

6. In figure 6, the authors distinguish between JAK1 mutant and WT cell by surface expression of HLA-DR. From the figure it seems that HLA-DR is a good marker for tumor cells with intact Interferon signaling. Can this be true also for the other cell lines described in the manuscript? For instance, is it possible that the ~30% UKE-mel-40c cells which do not upregulate CD54 and PD-L1 are also HLA-DR lo compared to the responding cells? Furthermore, the rational for selecting the cells based on HLA-DR is not clear, since HLA-DR is neither investigated nor discussed in the manuscript up to this point.

7. In figure 8 the authors show extensive mutational analysis of TCGA samples for genes in the Interferon- γ cascade (IFNGR1/2, JAK1/2, STAT1, IRF1). Do these mutations affect the survival curve of the patients, similar to the high vs. low levels of CD8A and IFNG?

8. In Table 1 the authors list the response of patients bearing mutation in the interferon pathway following anti-PD1 and anti-CTLA4, which for most cases was poor (partial response or progressive disease). It would be interesting to compare the poor response of these patients with the responses of the rest of the cohort. The data would predict that the response for the other patients would be better than the response of the patients in the table, and such clinical data is missing at the moment from the paper.

9. The TCGA subset used in the paper does not represent updated set of melanoma TCGA data available. There are now 479 samples which is double the number of samples used in the study.

10. Please provide a clear explanation/rational how the cell lines were picked for the functional studies including mutational background.

11. The authors show mutational analysis of TCGA samples for genes in the Interferon- γ cascade (IFNGR1/2, JAK1/2, STAT1, IRF1). Is there mutual exclusivity between the genes mentioned above? please provide a p value for this calculation. Do those genes correlate with survival?

12. Did the author check the co-ocurance between IFNG and CD8A? Please provide p values.

Minor comments:

1. Figure 1D- the figure show HLA class II staining, however HLA class II is not referred in the text and is also not shown elsewhere in the figure and its significance in the figure is not explained.

2. Figure 1f, 2d, 6h- the authors should show the gates of annexin V and PI staining by flow cytometry plots, at least as supplementary figures.

3. Figure 1g- in the histograms of HLA class I, CD54 and PD-L1 there are no numbers indicating the fluorescence intensity, as in the other histograms throughout the paper.

4. Figure 4c- in Ma-Mel-102, pSTAT1 is visible but STAT1 is completely missing, therefore it is difficult to estimate to what extent STAT1 phosphorylation is dampened.

5. In page 10- the authors write "As shown in Fig. 5a, metastasis Ma-Mel-36 developed after the patient had been treated with recombinant IFN α and a combination of dacarbazine/IL2/IFN α " etc., they must mean Fig 6a.

6. Figure 3a- no statistical test was assigned to the survival plots.

Reviewer #2 (Remarks to the Author):

The authors describe several cases of patients with melanoma who, thorough the course of their cancer evolution, became resistant to immunotherapy mediated by inactivating mutations in the interferon gamma receptor signaling pathway, in particular in JAK1 or JAK2.

Major comments:

Figures 1 and 2 present two cases that do not seem to be from patients with acquired IFN-g resistance, as cell lines from biopsies in these patients were responsive to IFN-g in vitro. It is unclear why these two cases are relevant and require being presented in two figures. The case in

Figure 1 would have been of relevance if the PD-L1 negative variants would have been sorted and shown to have an interferon pathway alteration leading to lack of response to signaling.

The description of mutations in JAK1 and JAK2 in the cell lines in Figure 4 should include details on the significance of the mutation for the JAK kinase function. Are these homozygous or heterozygous mutations?

Why are the WBs lacking JAK1 and JAK2 protein assessment in these cell lines? If the mutations lead to lack of protein production or to protein degradation then it would be of relevance to show that the protein is or not expressed.

How could Ma-Mel-102 have pSTAT1 without STAT protein in WB?

Ma-Mel-85 is presented as being non-responsive to IFN-g but it seems to be very responsive to its expected downstream signaling.

Ma-Mel-36 with a JAK1 truncation would be anticipated to not respond to IFN-alpha, beta and gamma, not only to IFN-g. This should be tested. Do these cells also become resistant to IFN-alpha-induced growth arrest?

In page 12 the authors state that they reported on four cases with IFN-g resistant melanoma cells, but the derived cell lines Ma-Mel-54 and Ma-Mel-102 do not seem to be resistant, while Ma-Mel-36 and Ma-Mel-61 are indeed resistant.

Given the postulates of this article, it seems to be of relevance to experimentally test the hypothesis by generating CRISPR/Cas9 knock down sublines of a responsive baseline line to demonstrate that the genetic alterations are the cause of resistance to IFN-g.

The authors end the Results section stating that IFN-g signaling alterations predispose to innate or acquired resistance upon immune therapeutic pressure. However, it is unclear from Table 1 which cases the authors are referring to. Several patients with a CR or PR to anti-PD-1 or ipilimumab therapy are cited as having baseline IRF1, JAK1 or JAK2 mutations, and their response to therapy seems to be independent of these mutations.

Minor comments:

To better place this work in context, the Introduction or Discussion should include several recent published articles of relevance to this field from the laboratories of Padmanee Sharma and Harry Hollema.

The heading "HLA class I-low phenotype maintenance due to defective IFNg signaling" is confusing. Do the authors mean that IFNg does no longer lead to increase in MHC class I expression?

On the top of page 13 the authors refer to "a remarkable fraction of the biopsies...". How many?

On the top of page 14 the authors refer to "some of them clearly inactivating...". Which ones?

The article has several typographical errors, references to wrong figures and requires English editing.

Point-by-point response to the referees' comments

Reviewer #1 (Reviewer Comments to the Authors):

A number of patients do not benefit from immunotherapy despite its considerable potential. In the paper entitled "Acquired IFN resistance impairs anti-tumor immunity in melanoma", Antje Sucker and colleagues discuss the necessity of identifying resistance mechanisms for advancing treatment of melanoma. The authors demonstrate the evolution of IFN resistance in melanoma cells. They tested whether melanoma cells from patients responding to immunotherapy are sensitive to the anti-proliferative and pro-apoptotic effects of IFN and that continuous cytokine exposure selects for the outgrowth of IFN-resistant tumor subclones. Their results show that various treatments meant to increase CD8+ T cell anti-tumor activity, such as checkpoint blockade inhibitors (PD-1, CTLA4), mediate the positive selection of tumor subclones which are immune to CD8+ anti-tumor activity by resistance to interferon- γ signaling. Interferon- γ promotes apoptosis and arrest cell growth in tumor cells with intact Interferon- γ signaling, whereas tumor cells with mutations in interferon associated downstream genes, such as kinases JAK1 and JAK2 and the transcription factor STAT1, do not undergo these effects; in addition, tumor cells bearing mutations for these genes do not upregulate surface HLA class I and PD-L1 as a result of Interferon- γ stimulation. The interferon-resistance cells are positively selected by the enhanced CD8+ T cell activity as a result of the treatment, and may partially explain the relapse seen in patients undergoing immunotherapy.

Although the findings and conclusions are important and would interest the melanoma research community, the data shown in the paper is mostly not novel. Many of the findings, including the emergence of JAK mutated subclones following treatment and their resistance to Interferon- γ effects on cell growth, were already reported previously by Zaretski et al, NEJM 2016. Similar findings were also published recently by Gao et al, Cell 2016, which reported even further mutations in other interferon related genes. While the authors slightly expand the principle of Interferon- γ resistance to other genes in the interferon cascade, they bring little new concepts regarding the resistance mechanisms. Therefore the manuscript in its current form is not suitable for a publication in Nature Communications.

Authors' reply: We thank the reviewer for emphasizing the importance of our work and for her/his constructive comments. We have attempted to address each of the concerns as best possible and extended our experimental studies in the different patient models, now providing evidence for a novel IFN γ -related resistance mechanism. We demonstrate that JAK1/2-deficient melanoma cells give rise to T cell-resistant HLA class I-negative lesions by

silencing the expression of genes involved in antigen presentation and blocking their IFN γ -dependent re-expression. These results have been generated by deeper analyses in the patient model Ma-Mel-54 (JAK2 mutant) and by work on an additional metastasis of patient Ma-Mel-61 (JAK1 mutant). To the best of our knowledge this is the first study demonstrating that IFN γ -resistant JAK1/2-deficient melanoma cells progress to a ‘higher level’ of immunotherapy resistance.

Major comments by Reviewer 1:

1. Figure 1- the title of the figure suggests the outgrowth of interferon-resistance cells following anti-PD1 treatment. However in this figure the only indication of the resistant cells is the minor populations that did not upregulate CD54 and PDL1 (35% and 28% respectively). The authors did not check if these minor population phosphorylate STAT1 or undergo apoptosis as performed in figure 1E-F for the whole population, or as they did for the HLA-DR+ and HLA-DR- cells in figure 6. It is recommended that figure 1G should go to supplementary data, since it is not fully indicate Interferon- γ resistance.

Authors’ reply: As the reviewer pointed out correctly, we did not study the mechanisms abrogating CD54 and PD-L1 upregulation in response to IFN γ treatment in a minor population of UKE-Mel-40c cells. Studies on the molecular mechanisms were not feasible due to the slow proliferation of UKE-Mel-40c cells and loss of the subpopulation during continuous cell passage. Therefore, we have removed former Fig. 1g from the paper.

2. Figure 1 and Figure 2 are very similar. Both show responsiveness of tumor cell lines derived before and following treatment with checkpoint inhibitors, each from one individual patient. It is not clear what new data does figure 2 add on top of figure 1, other than different treatment course due to the different clinical history of the two patients.

Authors’ reply: We agree with the reviewer that the data presented in former Fig. 2 (patient model UKE-Mel-62) to a certain degree recapitulated the results presented for patient UKE-Mel-40 in Figure 1. We have thus removed the data on patient model UKE-Mel-62 from the manuscript.

3. Figure 3, showing favorable clinical outcomes of the presence of CD8+ T cells and poor outcomes in their absence, is reminiscent of previous data in the literature (Taylor et al, Journal of clinical oncology 2007; Azimi et al, Journal of clinical oncology 2012, to name a few). Therefore it is recommended to cite these publications and move the figure to the supplementary data.

Authors’ reply: As requested by the reviewer, the survival plot demonstrating an association between *CD8A* mRNA expression and favorable clinical outcome is now shown as supplementary Fig. 1c. New

survival plots emphasizing the association of active IFN γ signaling with improved overall survival are now presented in Fig. 1g.

4. In figure 4, the author's show that Ma-Mel-102 cells are not responding properly to interferon, however they do not show ablation of the pro-apoptotic/ anti-proliferative effect of interferon. Thus, a functional readout for the STAT1 mutation is lacking. Similarly, a functional readout is missing in figure 5.

Authors' reply: We agree with the reviewer that data showing protection of tumor cells from anti-tumor IFN γ activity by acquisition of specific genetic alterations should be presented. In addition to the JAK1-deficient models (Ma-Mel-36, Ma-Mel-61g) we now provide these data for the JAK2-deficient patient model Ma-Mel-54 (Fig. 4h). In contrast to the JAK1/2-deficient cells, IFN γ signaling is strongly impaired but still active at a low level in Ma-Mel-102 cells. Therefore and for reasons of poor transfection efficiency, we did not include Ma-Mel-102 cells in our functional assays.

5. In figure 5, Ma-Mel-54a cells are HLA class I low, however Ma-Mel-54b do express HLA class I levels which are comparable with other cell lines described in the paper. The two cell lines have the exact JAK2 mutation (Q959P), however the difference in HLA class I expression is not explained. This is an important point since the authors claim that the JAK2 mutation keep HLA class I levels low. Figure 5D suggest that low HLA class I levels can be restored in a manner that is independent of interferon- γ , this should at least be discussed.

Authors' reply: We absolutely agree with the reviewer and studied the mechanisms leading to the HLA class I-low phenotype of Ma-Me-54a cells. By qRT-PCR analyses we demonstrate a coordinated downregulation of genes involved in antigen presentation in Ma-Mel-54a compared to Ma-Mel-54b cells (Fig. 4i). Expression of these genes can be induced in IFN γ -treated Ma-Mel-54a-JAK2 transfectants (Fig. 4j). Thus, JAK2 deficiency allows Ma-Mel-54a cells to conserve their HLA class I-low phenotype in the presence of IFN γ . Similar results were obtained for the JAK1-deficient Ma-Mel-61h cells (Fig. 6). These cells completely lack expression of HLA class I surface molecules that could only be induced by IFN γ in Ma-Mel-61h-JAK1 transfectants. Overall the data obtained in these two patient models indicate that JAK1/2-deficient melanoma cells give rise to T cell-resistant HLA class I-negative lesions by silencing the expression of genes involved in antigen presentation and blocking their IFN γ -dependent re-expression.

6. In figure 6, the authors distinguish between JAK1 mutant and WT cell by surface expression of HLA-DR. From the figure it seems that HLA-DR is a good marker for tumor cells with intact Interferon signaling. Can this be true also for the other cell lines described in the manuscript? For instance, is it

possible that the ~30% UKE-mel-40c cells which do not upregulate CD54 and PD-L1 are also HLA-DR lo compared to the responding cells? Furthermore, the rationale for selecting the cells based on HLA-DR is not clear, since HLA-DR is neither investigated nor discussed in the manuscript up to this point.

Authors' reply: As correctly indicated by the reviewer, we did not clarify the role of HLA-DR in our studies. Of the different IFN γ -sensitive cell lines cultured in our laboratory some show constitutive HLA-DR expression while others are negative but become strongly positive in the presence of IFN γ . This is also the case for the IFN γ -sensitive Ma-Mel-36 subpopulation (Fig. 3b), allowing us to apply HLA-DR as a marker to separate IFN γ -sensitive and IFN γ -resistant Ma-Mel-36 cells which we have now renamed Ma-Mel-36_sens and Ma-Mel-36_res cells, respectively.

7. In figure 8 the authors show extensive mutational analysis of TCGA samples for genes in the Interferon- γ cascade (IFNGR1/2, JAK1/2, STAT1, IRF1). Do these mutations affect the survival curve of the patients, similar to the high vs. low levels of CD8A and IFNG?

Authors' reply: We agree with the reviewer that it is important to define the impact of alterations in the IFN γ cascade on patient survival. Our analyses revealed that truncating mutations, homozygous deletions and low protein levels in IFNGR1/2, JAK1/2, STAT1, IRF1 were associated with decreased patient survival, as shown in Fig. 7c.

8. In Table 1 the authors list the response of patients bearing mutation in the interferon pathway following anti-PD1 and anti-CTLA4, which for most cases was poor (partial response or progressive disease). It would be interesting to compare the poor response of these patients with the responses of the rest of the cohort. The data would predict that the response for the other patients would be better than the response of the patients in the table, and such clinical data is missing at the moment from the paper.

Authors' reply: According to the reviewer's suggestion we analyzed the group of patients with and without mutations for differences in their "best clinical response". However, there is no significant difference between both groups, as shown in supplementary Table 2. Larger patient cohorts will need to be analyzed, which are however not yet available.

9. The TCGA subset used in the paper does not represent updated set of melanoma TCGA data available. There are now 479 samples which is double the number of samples used in the study.

Authors' reply: The reviewer correctly points out that the TCGA melanoma dataset includes sample data from 479 patients. However, complete information on mutations including SNV, Indels and CNV is available only for samples from 287 patients. Data from the 287 samples

were applied for analysis where complete information on mutations (SNV), Indels and CNV was required, including the calculation of mutation co-occurrence (supplementary Fig. 5). We have clarified this in the result section.

10. Please provide a clear explanation/rational how the cell lines were picked for the functional studies including mutational background.

Authors' reply: We have chosen our cell lines based on the following criteria: cell lines with heterozygous mutations (Ma-Mel-53, Ma-Mel-85) still showed intact IFN γ signaling and were not chosen for further functional studies. Of the cells with homozygous mutations, the *STAT1* mutant cell line was not used due to residual IFN γ signaling. Thus, we focused our functional assays on the homozygous *JAK1/2* mutant cells showing complete IFN γ resistance. We have clarified this in the result section.

11. The authors show mutational analysis of TCGA samples for genes in the Interferon- γ cascade (*IFNGR1/2*, *JAK1/2*, *STAT1*, *IRF1*). Is there mutual exclusivity between the genes mentioned above? please provide a p value for this calculation. Do those genes correlate with survival?

Authors' reply: According to the reviewer's suggestion, we studied mutations in genes of the IFN γ cascade (*IFNGR1/2*, *JAK1/2*, *STAT1*, *IRF1*) for their co-occurrence. This analyses was performed on the TCGA melanoma dataset (n=287) for which complete information on mutations and CNV is available. Only for *JAK1* and *STAT1* mutations was a tendency for co-occurrence identified (p=0.008, log OR>3). This is however based on only 4 *STAT1* mutations of which two co-occurred with two out of 11 *JAK1* mutations (supplementary Fig. 5). This result suggests that in general, mutations in the IFN γ cascade are mutually exclusive, as suggested also by Fig. 7a.

Furthermore, we demonstrate that expression of *STAT1* and *IRF1* mRNA, indicating active IFN γ signaling, are strongly correlated with improved survival, as shown in Fig. 1g.

12. Did the author check the co-occurrence between *IFNG* and *CD8A*? Please provide p values.

Authors' reply: As presented in the first version of the manuscript, TCGA melanoma sample data was assessed for the co-expression of *CD8A* and *IFNG* mRNA, finding a strong correlation (Pearson correlation 0.91, n=479, p< 0.00001) (supplementary Fig. 1d). Based on the reviewer's suggestion we in addition analyzed *IFNG* and *CD8A* expression levels in tumors with and without mutations. Applying the wilcoxon rank test we could not detect an association between *IFNG/CD8A* mRNA levels and mutations in genes of the IFN γ signaling pathway (supplementary Fig. 6).

Minor comments:

1. Figure 1D- the figure show HLA class II staining, however HLA class II is not referred in the text and is also not shown elsewhere in the figure and its significance in the figure is not explained.

Authors' reply: We stained the tissue for HLA class II expression since it is a known IFN γ -sensitive marker, to indicate regions of active IFN γ signaling. This has now been indicated in the result section (page 5).

2. Figure 1f, 2d, 6h- the authors should show the gates of annexin V and PI staining by flow cytometry plots, at least as supplementary figures.

Authors' reply: As requested by the reviewer, representative dot plots of AnnexinV/PI-stained cells are now presented as supplementary Figs. 1b, 3e and 3f.

3. Figure 1g- in the histograms of HLA class I, CD54 and PD-L1 there are no numbers indicating the fluorescence intensity, as in the other histograms throughout the paper.

Authors' reply: As indicated above, results presented in former Fig. 1g have been removed from the manuscript.

4. Figure 4c- in Ma-Mel-102, pSTAT1 is visible but STAT1 is completely missing, therefore it is difficult to estimate to what extent STAT1 phosphorylation is dampened.

Authors' reply: To clarify the results we obtained for Ma-Mel-102 cells we now present Western blot data from long-term exposed Xray films, in which signals for STAT1 and pSTAT1 become visible (Fig. 2). However, these signals are still weak compared to those obtained for Ma-Mel-85 cells analyzed in parallel, indicating impaired IFN γ signaling in Ma-Mel-102 cells.

5. In page 10- the authors write "As shown in Fig. 5a, metastasis Ma-Mel-36 developed after the patient had been treated with recombinant IFN α and a combination of dacarbazine/IL2/IFN α " etc., they must mean Fig 6a.

Authors' reply: We thank the reviewer for pointing this out, the error was corrected.

6. Figure 3a- no statistical test was assigned to the survival plots.

Authors' reply: Information about the statistical analyses has now been included, presented in Fig. 1g and supplementary Figs. 1c and 1d of the manuscript.

Reviewer #2 (Reviewer Comments to the Authors):

The authors describe several cases of patients with melanoma who, thorough the course of their cancer evolution, became resistant to immunotherapy mediated by inactivating mutations in the interferon gamma receptor signaling pathway, in particular in JAK1 or JAK2.

Authors' reply: We thank the reviewer for his/her work on our manuscript and have attempted to address each of the constructive comments as best possible.

Major comments:

1. Figures 1 and 2 present two cases that do not seem to be from patients with acquired IFN-g resistance, as cell lines from biopsies in these patients were responsive to IFN-g in vitro. It is unclear why these two cases are relevant and require being presented in two figures. The case in Figure 1 would have been of relevance if the PD-L1 negative variants would have been sorted and shown to have an interferon pathway alteration leading to lack of response to signaling.

Authors' reply: The former figures 1 and 2 demonstrated that tumor cells from anti-PD1 responders are highly sensitive to the anti-proliferative/pro-apoptotic effects of IFN γ . These results emphasize that IFN γ activity might have contributed to therapy efficacy but also exerts a strong selective pressure on the tumor cells. As data presented for the anti-PD1 responder UKE-Mel-62 (former Fig. 2) to a certain degree recapitulated the results presented for patient UKE-Mel-40 in Figure 1, we removed the data on patient model UKE-Mel-62 from the manuscript.

2. The description of mutations in JAK1 and JAK2 in the cell lines in Figure 4 should include details on the significance of the mutation for the JAK kinase function. Are these homozygous or heterozygous mutations?

Authors' reply: According to the reviewer's suggestion, we have included additional information on the allelic status of the mutations in the different cell lines, now presented in Table 1.

3. Why are the WBs lacking JAK1 and JAK2 protein assessment in these cell lines? If the mutations lead to lack of protein production or to protein degradation then it would be of relevance to show that the protein is or not expressed.

Authors' reply: We absolutely agree with the reviewer and performed JAK1/2 Western blots for mutant Ma-Mel-36, Ma-Mel-54 and Ma-Mel-61 cells as presented in Figs. 3, 4 and 5, respectively.

4. How could Ma-Mel-102 have pSTAT1 without STAT protein in WB?

Authors' reply: To clarify the results we obtained for Ma-Mel-102 cells we now present Western blot data from long-term exposed Xray films, in which signals for STAT1 and pSTAT1 become visible (Fig. 2). However, these signals are still weak compared to those obtained for Ma-Mel-85 cells analyzed in parallel, indicating impaired IFN γ signaling in Ma-Mel-102 cells.

5. Ma-Mel-85 is presented as being non-responsive to IFN-g but it seems to be very responsive to its expected downstream signaling.

Authors' reply: As correctly pointed out by the reviewer, Ma-Mel-85 cells are IFN γ -sensitive. We always classified Ma-Mel-85 cells as IFN γ -sensitive, but at some point our statements might have been misleading. We edited the text to prevent any misunderstanding (pages 6/7).

6. Ma-Mel-36 with a JAK1 truncation would be anticipated to not respond to IFN-alpha, beta and gamma, not only to IFN-g. This should be tested. Do these cells also become resistant to IFN-alpha-induced growth arrest?

Authors' reply: We agree with the reviewer that it is of importance also to study the response of Ma-Mel-36 cells to IFN α , as the patient received different IFN α -based therapies. We demonstrate that IFN γ -resistant Ma-Mel-36 cells (now termed Ma-Mel-36_res) are cross-resistant to IFN α (Fig. 3e). But in contrast to IFN γ , IFN α does not induce apoptosis in the sensitive Ma-Mel-36 (now termed Ma-Mel-36_sens) subpopulation (supplementary Fig. 3f).

7. In page 12 the authors state that they reported on four cases with IFN-g resistant melanoma cells, but the derived cell lines Ma-Mel-54 and Ma-Mel-102 do not seem to be resistant, while Ma-Mel-36 and Ma-Mel-61 are indeed resistant.

Authors' reply: Our data demonstrate that in addition to Ma-Mel-36 and Ma-Mel-61g/h cells also Ma-Mel-54 cells are IFN γ -resistant, showing no JAK/STAT/IRF signaling under cytokine treatment. Only Ma-Mel-54a-JAK2 transfectants respond to IFN γ (Fig. 4g, h, j, k). Indeed, Ma-Mel-102 STAT1 mutants show strongly impaired IFN γ signaling. This has now been clarified in the text (page 7).

8. Given the postulates of this article, it seems to be of relevance to experimentally test the hypothesis by generating CRISPR/Cas9 knock down sublines of a responsive baseline line to demonstrate that the genetic alterations are the cause of resistance to IFN-g.

Authors' reply: In different patient-derived JAK1/JAK2 mutant melanoma cells we can restore IFN γ signaling upon transfection of plasmids encoding wild-type JAK1/JAK2. The role of these genes in IFN γ signaling is well

established. Therefore, we think that CRISPR/CAS9 knock out of JAK1/JAK2 wild-type genes in melanoma cells is not essentially required.

9. The authors end the Results section stating that IFN-g signaling alterations predispose to innate or acquired resistance upon immune therapeutic pressure. However, it is unclear from Table 1 which cases the authors are referring to. Several patients with a CR or PR to anti-PD-1 or ipilimumab therapy are cited as having baseline IRF1, JAK1 or JAK2 mutations, and their response to therapy seems to be independent of these mutations.

Authors' reply: According to the reviewer's suggestion we modified our statement, now indicating which of the identified mutations are inactivating and relating this to the patient's therapy response (page 15).

Minor comments:

1. To better place this work in context, the Introduction or Discussion should include several recent published articles of relevance to this field from the laboratories of Padmanee Sharma and Harry Hollema.

Authors' reply: We agree with the reviewer regarding the importance of these published articles and have now referenced them in the discussion section of the manuscript.

2. The heading "HLA class I-low phenotype maintenance due to defective IFN γ signaling" is confusing. Do the authors mean that IFN γ does no longer lead to increase in MHC class I expression?

Authors' reply: According to the reviewers suggestion we modified the subheading as follows: "JAK2 deficiency blocks HLA class I upregulation by IFN γ " (page 9).

3. On the top of page 13 the authors refer to "a remarkable fraction of the biopsies...". How many?

Authors' reply: We have indicated the fraction of biopsies showing homozygous gene deletions (page 13).

4. On the top of page 14 the authors refer to "some of them clearly inactivating...". Which ones?

Authors' reply: We indicated the inactivating mutations in the manuscript text (page 15).

5. The article has several typographical errors, references to wrong figures and requires English editing.

Authors' reply: We thank the editor for this advice and edited/corrected the text.

Reviewers' comments:

Reviewer #1 (Remarks to the Author):

Sucker et al have revised their manuscript according to my comments. While they did add significant new data, I find the paper hard to follow, as the message of the paper is unclear. The main weakness of the paper is in the context of its novelty in the field. The main claim of the authors is that tumor cells develop interferon gamma resistance following immunotherapy, but this message is not clearly demonstrated in the manuscript, and crucial data are missing to support it. In addition the paper is too long and includes repetitive data derived from many cell lines which differ little from one another.

Major points

1. I do not understand what the data of patient UKE-mel-40 contributes to the paper, since it is not stated that this cell line has any mutation in any of the interferon gamma associated genes. Figure 1G should be shown alongside other Kaplan-Meier curves shown in Figure 7.
2. Figure 2 can be used as a better starting point for the paper, however without the functional assay of cell death I find it incomplete and unsatisfactory, as figure 3 delivers basically the same message (comparison of JAK1 mutant vs. WT) in a better fashion.
3. The main claim of the paper is the progression of the interferon resistance. The authors use different cell lines taken from Ma-Mel-61 patient and test for their interferon signaling. They show that the ma-mel-61g cell line is JAK1 mutated and does not respond to interferon, unlike the ma-mel-61b cell line which is derived from a previous metastasis. However, whether ma-mel-61b is JAK1 mutated or WT is unknown. This is a crucial and important point, and it must be shown to prove that overtime, following therapy, the interferon resistance really does develop.
4. Figure 7C, which shows a Kaplan-Meier curve for combined mutations in the interferon pathway, is very reminiscent to the one shown in Gao et al, Cell 2016, figure 1D. This again questions the paper's novelty.

Minor points

1. Table 1- Ma-mel-61 does not appear in the table.
2. The western blots of figure 2b need to be quantified with statistics.
3. Figure 1g- please state p value, currently it is just 0.

Reviewer #2 (Remarks to the Author):

This resubmission addresses the majority of the concerns from the prior submission. It includes new relevant data and reorganization of the prior data to present the story more logically, which makes it a highly compelling, timely and interesting article.

Despite the rapid publication of several relevant articles in this same field, I believe that this article provides novel and highly supportive data portraying how genetic loss of function mutations in the interferon receptor signaling pathway are of key importance for cancer cells to escape immune attack and become resistant to anti-PD-1 therapy.

Major points:

Two other important and highly related articles have been published since the last review, Shin et al. (Cancer Discovery) showing that pre-existing JAK1 or JAK2 mutations in melanoma and colon cancer lead to insensitivity to interferon gamma and result in lack of response to anti-PD-1 therapy despite those tumors having a very high mutational load. Another article by Benci et al (Cell) has a slightly more complicated story where chronic interferon gamma exposure resulted in immune escape, and this could be blocked by inhibiting interferon gamma signaling.

Point-by-point response to the referees' comments

Reviewer #1 (Reviewer Comments to the Authors):

Sucker et al have revised their manuscript according to my comments. While they did add significant new data, I find the paper hard to follow, as the message of the paper is unclear. The main weakness of the paper is in the context of its novelty in the field. The main claim of the authors is that tumor cells develop interferon gamma resistance following immunotherapy, but this message is not clearly demonstrated in the manuscript, and crucial data are missing to support it. In addition the paper is too long and includes repetitive data derived from many cell lines which differ little from one another.

Authors' reply: We thank the reviewer for emphasizing the significance of the new data we added to our manuscript. In order to shorten the manuscript some data was removed and other data included in the supplements, as indicated below.

Major comments by Reviewer 1:

1. I do not understand what the data of patient UKE-mel-40 contributes to the paper, since it is not stated that this cell line has any mutation in any of the interferon gamma associated genes. Figure 1G should be shown alongside other Kaplan-Meier curves shown in Figure 7.

Authors' reply: As suggested by the reviewer, data on the patient model UKE-Mel-40 was removed from the manuscript. The data from former Figure 1G was introduced into a new Figure 6, together with the Kaplan-Meier curves from former Figure 7.

2. Figure 2 can be used as a better starting point for the paper, however without the functional assay of cell death I find it incomplete and unsatisfactory, as figure 3 delivers basically the same message (comparison of JAK1 mutant vs. WT) in a better fashion.

Authors' reply: We agree with the reviewer that former Figure 3 is a good starting point for the manuscript. Data from former Figure 2 was included in the supplements (now supplementary Figure 1).

3. The main claim of the paper is the progression of the interferon resistance. The authors use different cell lines taken from Ma-Mel-61 patient and test for their interferon signaling. They show that the ma-mel-61g cell line is JAK1 mutated and does not respond to interferon, unlike the ma-mel-61b cell line which is derived from a previous metastasis. However, whether ma-mel-61b is JAK1 mutated or WT is unknown. This is a crucial and important point, and it must be shown to prove that overtime, following therapy, the interferon resistance really does develop.

Authors' reply: We agree with the reviewer that showing data demonstrating the presence or absence of *JAK1* mutations in all Ma-Mel-61 cell lines is relevant. Targeted sequencing demonstrated no *JAK1* mutations in Ma-Mel-61a, -61b, -61c and -61e cells. In contrast, the presented *JAK1* c.1798G>T, p.G600W mutation was detected in Ma-Mel-61g and Ma-Mel-61h cells. This information was included in Figure 3c, Figure 4a and supplementary Figure 3b.

4. Figure 7C, which shows a Kaplan-Meier curve for combined mutations in the interferon pathway, is very reminiscent to the one shown in Gao et al, Cell 2016, figure 1D. This again questions the paper's novelty.

Authors' reply: As correctly stated by the Reviewer, both analyses demonstrate similar results, even though our analysis is based on a different, larger set of gene alterations.

Minor comments:

1. Table 1- Ma-mel-61 does not appear in the table.

Authors' reply: Patient Ma-Mel-61 has been included in Table 1.

2. The western blots of figure 2b need to be quantified with statistics.

Authors' reply: From the Western blots of former Figure 2b (now supplementary Figure 1b) expression of IRF1, as a target of the JAK/STAT pathway and an indicator of active IFN γ signaling, has been quantified. The results of the quantification are presented in supplementary Figure 1c.

3. Figure 1g- please state p value, currently it is just 0.

Authors' reply: The p value for each Kaplan Meier curve has been included.

Reviewer #2 (Reviewer Comments to the Authors):

This resubmission addresses the majority of the concerns from the prior submission. It includes new relevant data and reorganization of the prior data to present the story more logically, which makes it a highly compelling, timely and interesting article.

Despite the rapid publication of several relevant articles in this same field, I believe that this article provides novel and highly supportive data portraying how genetic loss of function mutations in the interferon receptor signaling

pathway are of key importance for cancer cells to escape immune attack and become resistant to anti-PD-1 therapy.

Authors' reply: We thank the reviewer for her/his positive comments on our manuscript and for highlighting its relevance.

Major comments:

1. Two other important and highly related articles have been published since the last review, Shin et al. (Cancer Discovery) showing that pre-existing JAK1 or JAK2 mutations in melanoma and colon cancer lead to insensitivity to interferon gamma and result in lack of response to anti-PD-1 therapy despite those tumors having a very high mutational load. Another article by Benci et al (Cell) has a slightly more complicated story where chronic interferon gamma exposure resulted in immune escape, and this could be blocked by inhibiting interferon gamma signaling.

Authors' reply: We agree with the reviewer these recently published articles are highly relevant and are now referenced in the discussion section of the manuscript.

REVIEWERS' COMMENTS:

Reviewer #1 (Remarks to the Author):

Sucker et al have revised their manuscript according to my comments. From a technical point of view, the flow of the paper is now much improved by shortening and removal of unnecessary data, and the missing crucial data (the sequencing of Ma-Mel-61b, which is WT, compared with Ma-Mel-61g which is JAK1 mutated) was added. However, the data and interpretation contributes does not contribute novel insights to the field, given similar previous reports.

NCOMMS-16-21346C

Response of the authors to the comment of Reviewer 1

Reviewer #1 (Reviewer Comments to the Authors):

Sucker et al have revised their manuscript according to my comments. From a technical point of view, the flow of the paper is now much improved by shortening and removal of unnecessary data, and the missing crucial data (the sequencing of Ma-Mel-61b, which is WT, compared with Ma-Mel-61g which is JAK1 mutated) was added. However, the data and interpretation contributes does not contribute novel insights to the field, given similar previous reports.

Authors' reply: We thank the reviewer for emphasizing the improvement of the revised manuscript version. Our data demonstrate the genetic evolution of IFN γ resistance based on chromosomal alterations and subsequent inactivating mutations in the course of disease progression in different patient models. In addition we demonstrate that IFN γ -resistant melanoma cells can give rise to HLA class I-negative lesions by silencing the expression of genes involved in antigen presentation and blocking their IFN γ -dependent re-expression. Such lesions are completely resistant to CD8⁺ T cells. These findings have not been demonstrated by previous studies.